# SWELL1 is a glucose sensor regulating β-cell excitability and systemic glycaemia

Chen Kang[1], Litao Xie[1], Susheel K. Gunasekar[1], Anil Mishra[1], Yanhui Zhang[1], Saachi Pai[1], Yiwen Gao[1], Ashutosh Kumar[1], Andrew W. Norris[2,3], Samuel B. Stephens[1,3] & Rajan Sah[1,3,4]

Insulin secretion is initiated by activation of voltage-gated $Ca^{2+}$ channels (VGCC) to trigger $Ca^{2+}$-mediated insulin vesicle fusion with the β-cell plasma membrane. The firing of VGCC requires β-cell membrane depolarization, which is regulated by a balance of depolarizing and hyperpolarizing ionic currents. Here, we show that SWELL1 mediates a swell-activated, depolarizing chloride current ($I_{Cl,SWELL}$) in both murine and human β-cells. Hypotonic and glucose-stimulated β-cell swelling activates SWELL1-mediated $I_{Cl,SWELL}$ and this contributes to membrane depolarization and activation of VGCC-dependent intracellular calcium signaling. SWELL1 depletion in MIN6 cells and islets significantly impairs glucose-stimulated insulin secretion. Tamoxifen-inducible β-cell-targeted *Swell1* KO mice have normal fasting serum glucose and insulin levels but impaired glucose-stimulated insulin secretion and glucose tolerance; and this is further exacerbated in mild obesity. Our results reveal that β-cell SWELL1 modulates insulin secretion and systemic glycaemia by linking glucose-mediated β-cell swelling to membrane depolarization and activation of VGCC-triggered calcium signaling.

[1] Department of Internal Medicine, Division of Cardiovascular Medicine, University of Iowa, Carver College of Medicine, Iowa City, IA 52242, USA. [2] Department of Pediatrics, University of Iowa, Carver College of Medicine, Iowa City, IA 52242, USA. [3] Fraternal Order of the Eagles Diabetes Research Center, Iowa City, IA 52242, USA. [4] Abboud Cardiovascular Research Center, University of Iowa Carver College of Medicine, Iowa City, IA 52242, USA. Correspondence and requests for materials should be addressed to R.S. (email: rajan-sah@uiowa.edu)

Type 2 diabetes is characterized by both a loss of insulin sensitivity and, ultimately, a relative loss of insulin secretion from the pancreatic β-cell[1–3]. Accordingly, therapeutic strategies for the treatment of diabetes aim to improve insulin sensitivity (thiazolidinediones) or augment insulin secretion from the pancreatic β-cell (sulphonylurea receptor inhibitors). Insulin secretion from the pancreatic β-cell is triggered by $Ca^{2+}$ influx through voltage-gated $Ca^{2+}$ channels (VGCC) to promote insulin vesicle fusion with the β-cell plasma membrane. The firing of VGCC depends on the β-cell membrane potential, which is in turn mediated by the balance of depolarizing (excitatory) and hyperpolarizing (inhibitory) ionic currents[4]—thus, the β-cell membrane potential is a critical regulator of insulin secretion. Hyperpolarizing, inhibitory potassium currents have been extensively studied, including $I_{KATP}$[5], delayed rectifier $K^+$ currents[4,6], and more recently TASK-1[7] and TALK-1[8] potassium channels. Indeed, a cornerstone of current diabetes pharmacotherapy, the sulphonylurea receptor inhibitors (i.e., glibenclamide/glyburide), is aimed at antagonizing the hyperpolarizing $I_{K,ATP}$ current to facilitate β-cell depolarization and thereby potentiate insulin secretion. While much attention has focused on these inhibitory hyperpolarizing currents, there is little knowledge about the excitatory currents required to depolarize the β-cell in the first place, including the molecular identity of these currents[4], with the exception of emerging data demonstrating that transient receptor potential (TRP) channels may contribute to β-cell excitability[9–12] in mice.

There is a longstanding hypothesis, first proposed in the 1990s, that an elusive chloride ($Cl^−$) conductance known as the volume regulatory anion current (VRAC) or swell-activated $Cl^−$ current ($I_{Cl,SWELL}$) is responsible for an important glucose sensitive, swell-activated depolarizing current that is required for β-cell depolarization and subsequent activation of VGCC-mediated $Ca^{2+}$ signaling, insulin vesicle fusion and insulin secretion[13–20]. These studies and others propose that β-cell depolarization requires not only the release of a "brake," mediated by $I_{KATP}$, but also activation of an undiscovered swell-activated "accelerator" to drive the β-cell membrane potential to the threshold potential for VGCC activation required for insulin secretion.

In the current study, we combine β-cell patch clamp with shRNA- and CRISPR/cas9-mediated gene silencing, to show for the first time that the gene Lrrc8a, a member of the leucine-rich repeat (LRR) containing proteins[21,22] (Swell1) is required for this prominent swell-activated chloride current in MIN6 β-cells, and in mouse and human primary β-cells. SWELL1 forms a heteromultimeric channel with LRRC8b-e[21–23] and is responsible for VRAC and $I_{Cl,SWELL}$ in cell lines. SWELL1 and associated LRRC8b-e are broadly expressed[21,24], and although principally thought of as volume regulatory ion channels, the physiological function of SWELL1–LRRC8 channels remain unknown. We show that SWELL1 is required for normal swell and glucose-stimulated β-cell membrane depolarization, $Ca^{2+}$ signaling, insulin secretion, and systemic glucose homeostasis in response to a glucose load. These data highlight SWELL1-mediated "swell-secretion" coupling as required for glucose-stimulated insulin secretion (GSIS) and for regulation of systemic glycaemia.

## Results

**SWELL1 mediates swell-activated $I_{Cl,SWELL}$ in β-cells.** Recent studies have identified SWELL1 as a required component of a swell-activated $Cl^−$ current $I_{Cl,SWELL}$ or VRAC in common cell lines[21,22], forming multimeric channels with LRRC8b-e[23]. To determine if SWELL1 is also required for $I_{Cl,SWELL}$ in pancreatic β-cells, we adenovirally transduced mouse insulinoma (MIN6) cells with either an shRNA-directed against Swell1 (Ad-U6-

shswell1-mCherry; Fig. 1a) or a scrambled shRNA control (Ad-U6-shSCR-mCherry). We observe robust knockdown of SWELL1 protein (Fig. 1b and Supplementary Fig. 6a) and a significant reduction in hypotonic swell-activated $I_{Cl,SWELL}$ in Ad-shSwell1 relative to Ad-shSCR-transduced MIN6 cells (Fig. 1c, d). To determine whether SWELL1 is also required for $I_{Cl,SWELL}$ in mouse primary β-cells, we isolated islets from Swell1 floxed mice ($Swell1^{fl/fl}$)[25] and transduced them with an adenovirus-expressing GFP under control of a rat insulin promoter (Ad-RIP2-GFP) to allow positive identification of β-cells (GFP+cells). $Swell1^{fl/fl}$ islets were further treated with either an adenovirus-expressing Cre-mCherry to induce Cre-mediated excision of the floxed Swell1 allele or a control virus expressing mCherry alone (Supplementary Fig. 1a). By selecting GFP+/mCherry+ cells, we patch clamped either control WT β-cells ($Swell1^{fl/fl}$ β-cells) or Swell1 KO β-cells ($Swell1^{fl/fl}$/Cre β-cells; Fig. 1e). We find that WT β-cells express substantial swell-activated current that is entirely abolished upon Cre-mediated recombination in Swell1 KO β-cells (Fig. 1f–h). We next tested whether SWELL1 is also required for $I_{Cl,SWELL}$ in human β-cells and applied a similar approach. We transduced human islets with Ad-RIP2-GFP and Ad-shSwell1-mCherry or Ad-sh-SCR-mCherry (Supplementary Fig. 1b), in order to isolate and patch clamp human β-cells (GFP+) subjected to shRNA-mediated SWELL1 KD or to a scrambled control (GFP+/mCherry+; Fig. 1i). Similar to mouse β-cell recordings, we find that human β-cells also express significant SWELL1-mediated swell-activated current (Fig. 1j–l). Indeed, in all β-cells patch clamped, the reversal potential is ~−12 mV, which is near the reversal potential for $Cl^−$ under our recording conditions, consistent with SWELL1 mediating a swell-activated $Cl^−$ conductance in β-cells, as reported previously[13,16,17]. These data demonstrate that SWELL1 is required for VRAC or $I_{Cl,SWELL}$ in pancreatic β-cells.

**Extracellular glucose activates SWELL1-mediated $I_{Cl,SWELL}$ in β-cells.** Having established that SWELL1 is required for this previously enigmatic depolarizing swell-activated $Cl^−$ current[16,17] in MIN6 cells, and in both mouse and human primary β-cells, we asked whether glucose-mediated β-cell swelling[19] is sufficient to activate SWELL1-mediated $I_{Cl,SWELL}$. First, we measured β-cell size by light microscopy in WT and Swell1 KO/KD primary murine and human β-cells, respectively, in response to glucose-stimulated swelling (at 35–37 °C). WT murine β-cells swell 6.8 ± 1.6% in cross-sectional area upon perfusion of 16.7 mM glucose (from 1 mM basal glucose) and reach a maximum size at 12 min post glucose stimulation, followed by a reduction in β-cell size (Fig. 2a), consistent with regulatory volume decrease (RVD). In contrast, Swell1 KO murine β-cells swell monotonically to 8.2 ± 2.4% and exhibit no RVD (Fig. 2a). WT human β-cells show a similar trend, swelling 8.6 ± 3.5%, followed by RVD, whereas Swell1 KD human β-cells swell monotonically to 6.0 ± 1.5% (Supplementary Fig. 2a), and similar to Swell1 KO murine β-cells (Fig. 2a), fail to exhibit RVD. These data indicate that increases in glucose induce β-cell swelling and that SWELL1 is required for RVD in primary β-cells, as observed in cell lines[21,22,26]. Next, we applied the perforated patch clamp technique to primary β-cells at 35–37 °C in order to measure currents under the same conditions that induce glucose-mediated β-cell swelling. We find that increases in glucose (16.7 mM) activate an outwardly rectifying current in both mouse (Fig. 2b, d) and human (Supplementary Fig. 2b, c) β-cells. This outwardly rectifying glucose-activated β-cell current is blocked by the selective VRAC or $I_{Cl,SWELL}$ inhibitor, DCPIB (Fig. 2d and Supplementary Fig. 2b,c), and is absent in Swell1 KO murine β-cells (Fig. 2c, e, f). Importantly, the activation time-course of the glucose-stimulated SWELL1-

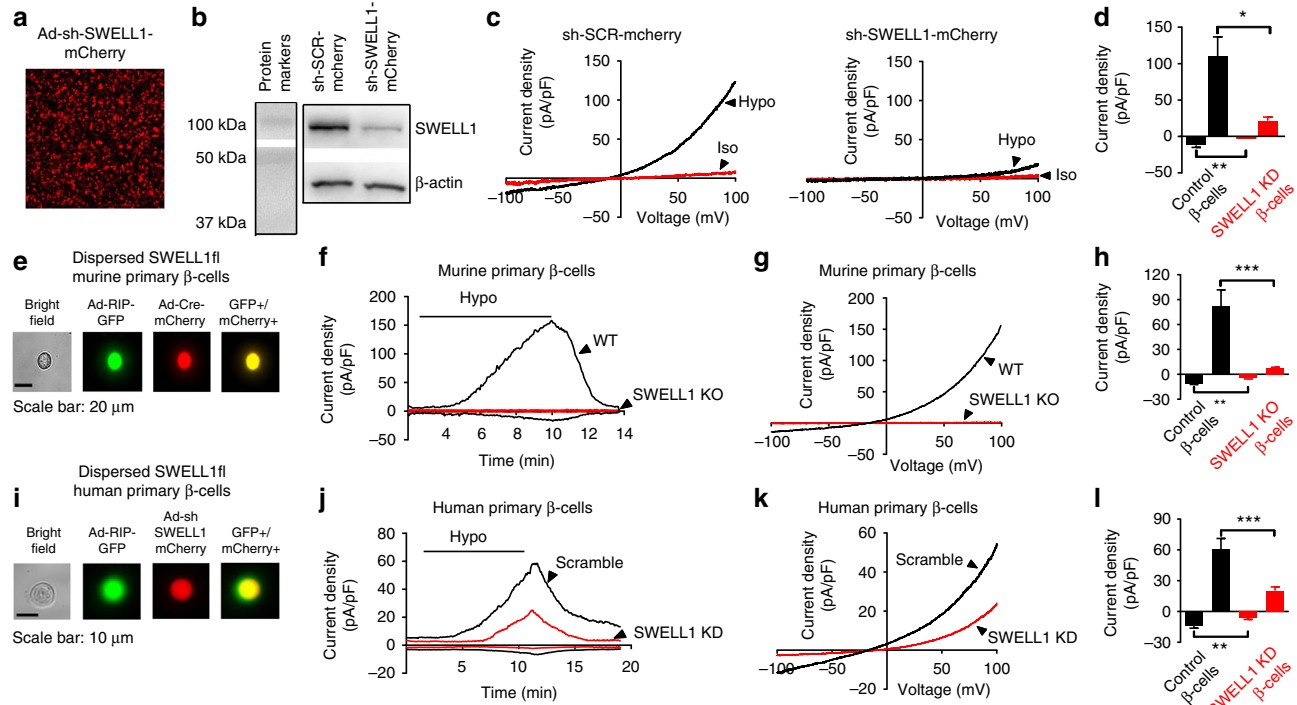

**Fig. 1** SWELL1 mediates $I_{Cl,SWELL}$ in MIN6 cells and primary mouse and human pancreatic β-cells. **a** mCherry fluorescence of the mouse insulinoma cell line (MIN6) transduced with an adenovirus expressing a short hairpin RNA directed to *Swell1* (sh*Swell1*-mCherry). **b** SWELL1 western blot in MIN6 cells transduced with sh*Swell1* compared to scrambled short hairpin RNA (shSCR). β-actin was used as loading control (Supplementary Fig. 6a for full blots). **c** Current–voltage relationship of $I_{Cl,SWELL}$ in MIN6 cells at baseline (red) and after hypotonic swelling (Hypo, 210 mOsm/kg, black) after adenoviral transduction with shSCR (left) and sh*Swell1* (right). **d** Mean current inward and outward densities at +100 and −100 mV ($n_{shSCR} = 3$ cells; $n_{shswell1} = 4$ cells). **e** Bright field, GFP, mCherry, and merged images of freshly dispersed primary β-cells from *Swell1$^{fl/fl}$* mouse islets co-transduced with Ad-RIP2-GFP and Ad-CMV-Cre-mCherry. Scale bar represents 20 µm. **f, g** Current–time relationship (**f**) and current–voltage relationship (**g**) of swell-activated $I_{Cl,SWELL}$ in wild-type (WT: Ad-CMV-mCherry/*Swell1$^{fl/fl}$*) and *Swell1* knockout (KO: Ad-CMV-Cre-mCherry/*Swell1$^{fl/fl}$*) mouse primary β-cells. **h** Mean current inward and outward densities at +100 and −100 mV ($n = 5$ cells, each group). **i** Bright field, GFP, mCherry, and merged images of freshly dispersed primary β-cells from human islets co-transduced with Ad-RIP2-GFP and Ad-sh*Swell1*-mCherry. Scale bar represents 10 µm. **j** Current–time relationship and **k** current–voltage relationship of swell-activated $I_{Cl,SWELL}$ in WT and SWELL1 knockdown primary human β-cells. **l** Mean current inward and outward densities at +100 and −100 mV ($n_{shSCR} = 5$ cells; $n_{shswell1} = 10$ cells). Ramp protocol is from +100 mV to −100 mV (ramp duration: 500 ms, holding potential: 0 mV). Data are shown as mean ± s.e.m. *$p < 0.05$; **$p < 0.01$; ***$p < 0.001$, unpaired *t*-test for **d**, **h**, **l**

mediated current either tracks or lags the latency of β-cell swelling in response to stimulatory glucose, consistent with a mechanism of glucose-mediated β-cell swell activation. Thus, SWELL1 mediates a glucose sensitive swell-activated Cl⁻ current in β-cells.

**SWELL1 depletion reduces β-cell membrane depolarization**. To determine whether the inward Cl⁻ current carried by SWELL1–LRRC8 channels is sufficient to depolarize the β-cell, we next measured β-cell membrane potential (MP) in murine and human WT and SWELL1-deficient β-cells upon hypotonic β-cell swelling (210 mOsm/kg) in current-clamp mode using whole-cell configuration (0 mM ATP$_i$; Fig. 3a–f). Under these conditions, K$_{ATP}$ channels will remain open while SWELL1-mediated $I_{Cl,SWELL}$ is selectively activated by hypotonic swelling. We find that β-cell resting MP is similar between WT and *Swell1* KO/KD β-cells (Fig. 3b, e) under basal conditions; however, the β-cell membrane depolarization rate (Fig. 3c, f) is significantly reduced 1.9-fold in *Swell1* KO murine β-cells and 2.5-fold in SWELL1-deficient human β-cells upon hypotonic swelling. These data confirm that hypotonic swell-activated SWELL1-mediated $I_{Cl,SWELL}$ contributes to β-cell membrane depolarization.

To determine the contributions of SWELL1-mediated $I_{Cl,SWELL}$ to glucose stimulation in β-cells, we next measured β-cell MP in

response to 16.7 mM glucose in WT and *Swell1* KO murine β-cells at 37 °C in perforated patch configuration (Fig. 3g), similar to recording conditions used to measure glucose-stimulated SWELL1 currents shown in Fig. 2. WT and *Swell1* KO β-cells have comparable resting MP (Fig. 3h) while glucose-stimulated β-cell membrane depolarization rate is significantly reduced 2.6-fold in *Swell1* KO β-cells (Fig. 3i). Collectively, these data show that SWELL1-mediated $I_{Cl,SWELL}$ contributes a significant glucose-stimulated depolarizing current in β-cells.

**β-cell SWELL1 is required for glucose and swelling-induced Ca²⁺ signaling**. Having established that SWELL1-mediated $I_{Cl,SWELL}$ contributes significantly to both glucose and hypotonic swell-activated β-cell membrane depolarization, we next sought to examine intracellular Ca²⁺ signaling in WT and SWELL1-deficient MIN6 β-cells, primary mouse and primary human β-cells in response to these stimuli. Using CRISPR/Cas9 technology, we generated multiple *Swell1* KO MIN6 cell lines (Supplementary Fig. 3), confirming *Swell1* gene disruption by PCR (Supplementary Fig. 3), SWELL1 protein deletion (Fig. 4a and Supplementary Fig. 6b) and ablation of SWELL1-mediated current (Fig. 4b) in these cells. We find that glucose-stimulated Ca²⁺ transients are entirely abolished in *Swell1* KO MIN6 compared to WT cells (Fig. 4c–d, f), despite preserved KCl (40 mM) stimulated Ca²⁺

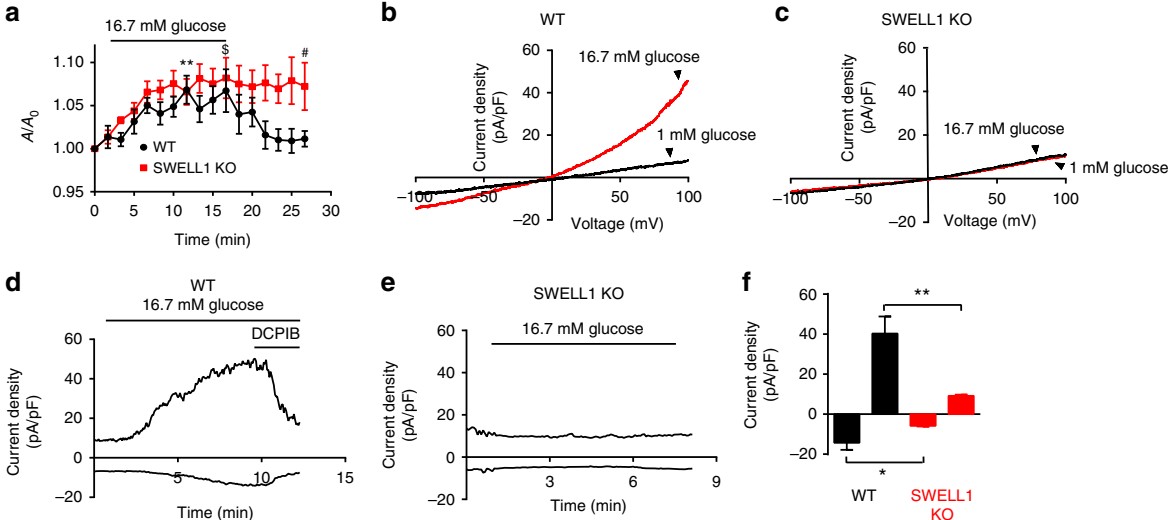

**Fig. 2** β-cell $I_{Cl,SWELL}$ is activated by physiological swelling in response to glucose stimulation. **a** Cross-sectional area of primary WT ($n = 12$ cells) and *Swell1* KO ($n = 7$ cells) murine β-cells in response to glucose stimulation (16.7 mM glucose). **b**, **c** Representative current–voltage relationship of $I_{Cl,SWELL}$ in **b** WT and **c** *Swell1* KO murine primary β-cell in response to 1 mM glucose (black trace) and 16.7 mM glucose (red trace). **d**, **e** Representative current–time relationship of $I_{Cl,SWELL}$ from **d** WT and **e** *Swell1* KO murine primary β-cell upon application of 16.7 mM glucose+DCPIB (10 μM, in WT only). **f** Mean current inward and outward densities at +100 and −100 mV (WT = 4 cells; *Swell1* KO = 5 cells). Recordings in **b**–**e** were all performed at 35–37 °C in perforated patch configuration. Ramp protocol is from +100 mV to −100 mV (ramp duration: 500 ms, holding potential: 0 mV). Data are shown as mean ± s.e.m. In **a**, \*\*$p < 0.01$ vs. 0 min in WT, paired *t*-test; $^{\$}p < 0.05$ vs. 0 min in *Swell1* KO, paired *t*-test; $^{\#}p < 0.05$ WT vs. *Swell1* KO, unpaired *t*-test. In **f**, \*$p < 0.05$; \*\*$p < 0.01$, unpaired *t*-test

transients (control for intact β-cell excitability). Co-application of a selective VGCC blocker nifedipine (10 μM) fully inhibits these glucose-stimulated $Ca^{2+}$ transients in WT MIN6 cells, consistent with a mechanism of membrane depolarization and VGCC activation (Fig. 4e, f).

As β-cells are also known to depolarize, fire $Ca^{2+}$ transients, and secrete insulin via a glucose-independent hypotonic swelling mechanism[16,20], we next examined swell-induced $Ca^{2+}$ signaling in β-cells in response to hypotonic stimulation (220 mOsm/kg) in the absence of glucose stimulation (0 mM glucose). These conditions are anticipated to selectively activate SWELL1-mediated $I_{Cl,SWELL}$ (Fig. 4b) while leaving $K_{ATP}$ channels open, thereby dissociating the contributions of SWELL1-mediated $I_{Cl,SWELL}$ activation from $K_{ATP}$ closure. We find that hypotonic swelling alone can trigger robust $Ca^{2+}$ transients in WT MIN6 cells (Fig. 4g, j) and these elevations in cytosolic $Ca^{2+}$ recover rapidly upon restoration of isotonic solution (Fig. 4g). In contrast, *Swell1* KO MIN6 cells do not exhibit hypotonic swelling-induced $Ca^{2+}$ transients (Fig. 4h, j), despite preserved KCl-stimulated $Ca^{2+}$ responses, consistent with SWELL1 mediating a swell-activated depolarizing current in β-cells (Figs. 3a–f and 4b). As with glucose-stimulated $Ca^{2+}$ signaling, we also find that hypotonic swelling triggered $Ca^{2+}$ transients are fully inhibited by VGCC blockade (Fig. 4i, j), implicating β-cell membrane depolarization followed by VGCC activation, as opposed to alternative hypo-osmotically activated $Ca^{2+}$ influx pathways, such as TRP channels[27].

To measure SWELL1-dependent $Ca^{2+}$ signaling in primary mouse and human β-cells, we generated adenoviruses expressing the genetically encoded $Ca^{2+}$-sensor GCaMP6s under control of the rat insulin promoter 1 (RIP1), either alone (Ad-RIP1-GCaMP6s), or in combination with Cre-recombinase (Ad-RIP1-Cre-P2A-GCaMP6s). This approach provides a robust β-cell-restricted fluorescent $Ca^{2+}$ sensor while simultaneously allowing for β-cell-targeted *Cre*-mediated *Swell1* deletion in cultured islets isolated from *Swell1*$^{fl/fl}$ mice (Fig. 5a). GCaMP6s $Ca^{2+}$ imaging reveals robust glucose-stimulated $Ca^{2+}$ transients in freshly

dissociated WT primary murine β-cells (Ad-RIP1-GCaMP6s/ *Swell1*$^{fl/fl}$; Fig. 5b, d) and these are significantly suppressed in *Swell1* KO β-cells (Ad-RIP1-Cre-P2A-GCaMP6s/*Swell1*$^{fl/fl}$; Fig. 5c, d), despite preserved KCl-stimulated $Ca^{2+}$ responses. We used a similar approach in human islets, whereby we co-transduced islets with Ad-RIP1-GCaMP6s (Fig. 5e) and either Ad-U6-sh*Swell1*-mCherry or Ad-U6-shSCR-mCherry. Upon islet dissociation, we imaged only double-labeled GCaMP6s +/mCherry+primary human β-cells. As with mouse primary β-cells, we observe robust glucose-stimulated $Ca^{2+}$ transients in Ad-shSCR-treated human primary β-cells (Fig. 5f, h) and this is markedly aborgated upon Ad-sh*Swell1*-mediated SWELL1 knockdown (Fig. 5g, h). Collectively, these data demonstrate that SWELL1 mediates a glucose and hypotonic swell-sensitive depolarizing $I_{Cl,SWELL}$ current that is necessary for normal β-cell depolarization and consequent intracellular $Ca^{2+}$ signaling.

**The balance of $I_{Cl,SWELL}$ and $I_{KATP}$ regulates β-cell $Ca^{2+}$ signaling.** Physiological increases in extracellular glucose and subsequent glucose metabolism induces both β-cell swelling[15] (Fig. 2a and Supplementary Fig. 2a) and increases β-cell cytoplasmic ATP/ADP ratio[2]. Therefore, glucose metabolism is predicted to concurrently activate depolarizing SWELL1-mediated $I_{Cl,SWELL}$ and deactivate hyperpolarizing $I_{KATP}$—suggesting that SWELL1 and $K_{ATP}$ coordinately regulate β-cell membrane depolarization. Indeed, application of 3-O-methyl-glucose, a non-metabolizable form of glucose, is ineffective at stimulating β-cell $Ca^{2+}$ transients (Fig. 6a)[19], however, it is also incapable of inducing β-cell swelling[19] and $K_{ATP}$ closure, as both require glucose metabolism[19]. To parse the relative contributions of depolarizing SWELL1-mediated $I_{Cl,SWELL}$ and hyperpolarizing $I_{KATP}$ to β-cell excitability, we measured intracellular $Ca^{2+}$ in WT and *Swell1* KO β-cells in response to selective modulators of either $K_{ATP}$ or $[Cl^-]_i$, the latter indirectly regulating $I_{Cl,SWELL}$. In pancreatic β-cells, $[Cl^-]_i$ is maintained relatively elevated (34–36 mM)[28,29] by $Na^+/K^+/Cl^-$ (NKCC1) cotransporters[30,31]. This

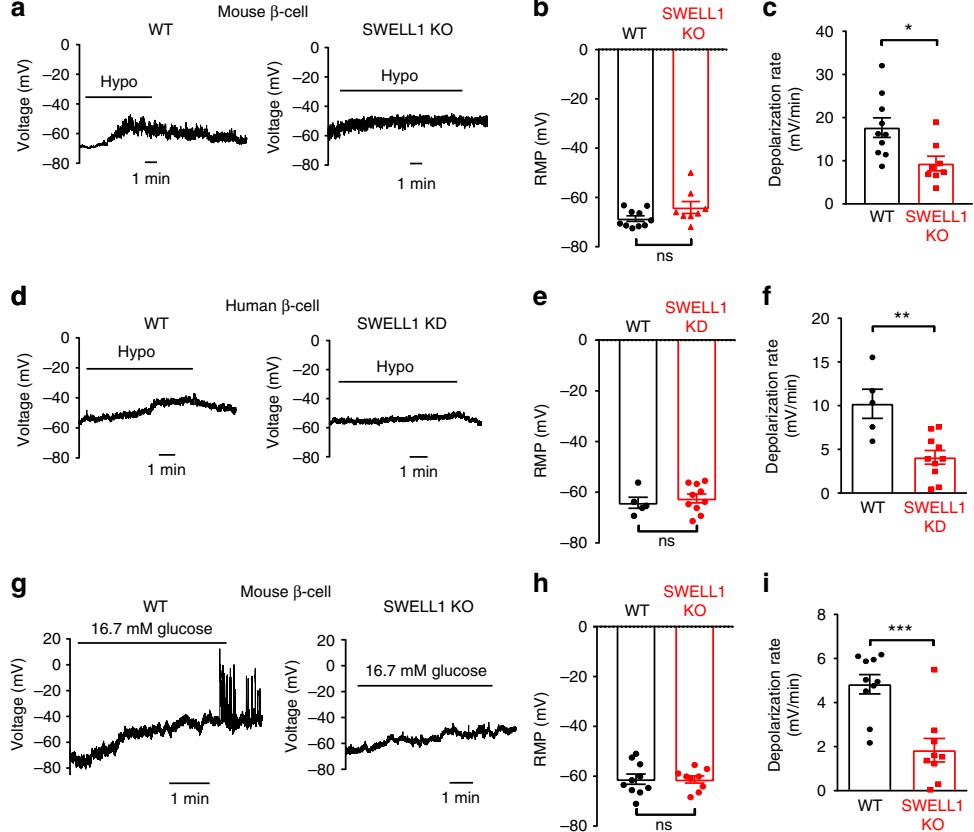

**Fig. 3** SWELL1 depletion reduces swell- and glucose-stimulated β-cell membrane depolarization. **a** β-cell membrane potential measured in current-clamp mode upon application of hypotonic solution (210 mOsm/kg) in WT (left) and *Swell1* KO (right) murine primary β-cells. **b** Resting membrane potential (RMP) and **c** membrane depolarization rate in WT ($n = 10$) and *Swell1* KO ($n = 8$) murine primary β-cells in response to hypotonic swelling. **d** β-cell membrane potential measured in current-clamp mode upon application of hypotonic solution (210 mOsm/kg) in WT (left) and SWELL1 KD (right) human primary β-cells. **e** RMP and **f** membrane depolarization rate in WT ($n = 5$) and SWELL1 KD ($n = 10$) human primary β-cells in response to hypotonic swelling. **g** Glucose (16.7 mM)-elicited membrane depolarization in isolated WT (left) and *Swell1* KO murine β-cells. **h** RMP and **i** membrane depolarization rate in WT ($n = 10$) and *Swell1* KO ($n = 9$) murine primary β-cells in response to 16.7 mM glucose stimulation from 1 mM basal glucose. All of the recordings above were performed at 35–37 °C in perforated patch configuration, current-clamp mode. Data are shown as mean ± s.e.m. *$p < 0.05$; **$p < 0.01$; ***$p < 0.001$, unpaired *t*-test. ns, not significant

generates a depolarizing $Cl^-$ current upon activation of a $Cl^-$ conductance (i.e., $Cl^-$ efflux via $I_{Cl,SWELL}$), since $E_{Cl-} = \sim -15$ mV[29,32], while resting β-cell membrane potential is $\sim -70$ mV. NKCC1 inhibition by bumetanide[33,34] reduces $[Cl^-]_i$, which drops $E_{Cl-}$ closer to −70 mV, thereby diminishing or abolishing the glucose-stimulated depolarizing current carried by $I_{Cl,SWELL}$. Indeed, we find that bumetanide (10 μM) fully inhibits glucose-stimulated (16.7 mM) $Ca^{2+}$ signaling in both WT MIN6 cells (Fig. 6b, c) and WT primary murine β-cells (Fig. 6d, e), to an extent comparable to *Swell1* KO β-cells (Fig. 5c, d). These data suggest that elevated $[Cl^-]_i$ maintained by β-cell NKCC1 activity is necessary for β-cell depolarization via a SWELL1-mediated glucose-stimulated depolarizing $Cl^-$ current.

To examine the contribution of $I_{KATP}$ to β-cell depolarization in WT and *Swell1* KO β-cells, we measured basal and glucose-stimulated intracellular $Ca^{2+}$ in response to the $K_{ATP}$ opener, diazoxide, and $K_{ATP}$ inhibitor, glibenclamide. $K_{ATP}$ activation (100 μM diazoxide) fully suppresses glucose-stimulated intracellular $Ca^{2+}$ in murine β-cells (Fig. 6f, g), without blocking $I_{Cl,SWELL}$ (Supplementary Fig. 4a, b), indicating that $K_{ATP}$ closure is necessary for glucose-stimulated β-cell excitation. Moreover, application of low-dose glibenclamide (0.25 nM), which is predicted to block ~25–30% of $K_{ATP}$ channels[35] (without activating $I_{Cl,SWELL}$, Supplementary Fig. 4c, d) is sufficient to stimulate WT but not *Swell1* KO β-cells (Fig. 6h–j). These data

suggests that, under basal conditions, there is a background of constitutively active SWELL1-mediated depolarizing $I_{Cl,SWELL}$ that is balanced by hyperpolarizing $I_{KATP}$ to maintain resting β-cell membrane potential. Indeed, basal $I_{Cl,SWELL}$ has been reported in neurons[36,37] and, importantly, has been measured in β-cells (NP = ~0.06, at 1 mM glucose)[38]. Near full $I_{KATP}$ inhibition with higher dose glibenclamide (10 nM, 0 mM glucose)[35] is capable of activating intracellular $Ca^{2+}$ in both WT and *Swell1* KO β-cells (Fig. 6k–m), just as robust $I_{Cl,SWELL}$ activation with hypotonic swelling (at 0 mM glucose) can overcome $I_{KATP}$ and trigger $Ca^{2+}$ transients (Fig. 4g) and insulin release from β-cells[16]. Taken together, these data support a model whereby SWELL1-mediated $I_{Cl,SWELL}$ and $I_{KATP}$ counterbalance each other to regulate glucose-stimulated β-cell $Ca^{2+}$ signaling.

**SWELL1 depletion selectively impairs glucose-mediated insulin secretion.** To determine the impact of SWELL1-dependent glucose-stimulated $Ca^{2+}$ signaling on insulin secretion in β-cells, we measured glucose-stimulated insulin secretion (GSIS) in WT and *Swell1* KO MIN6 cells. We find that the glucose-dependent (1, 5.5, and 30 mM) increase in insulin secretion in WT MIN6 cells is significantly diminished in *Swell1* KO MIN6 cells (Fig. 7a), particularly at higher glucose concentration (30 mM), despite no change in total insulin content (Fig. 7b). We next isolated islets

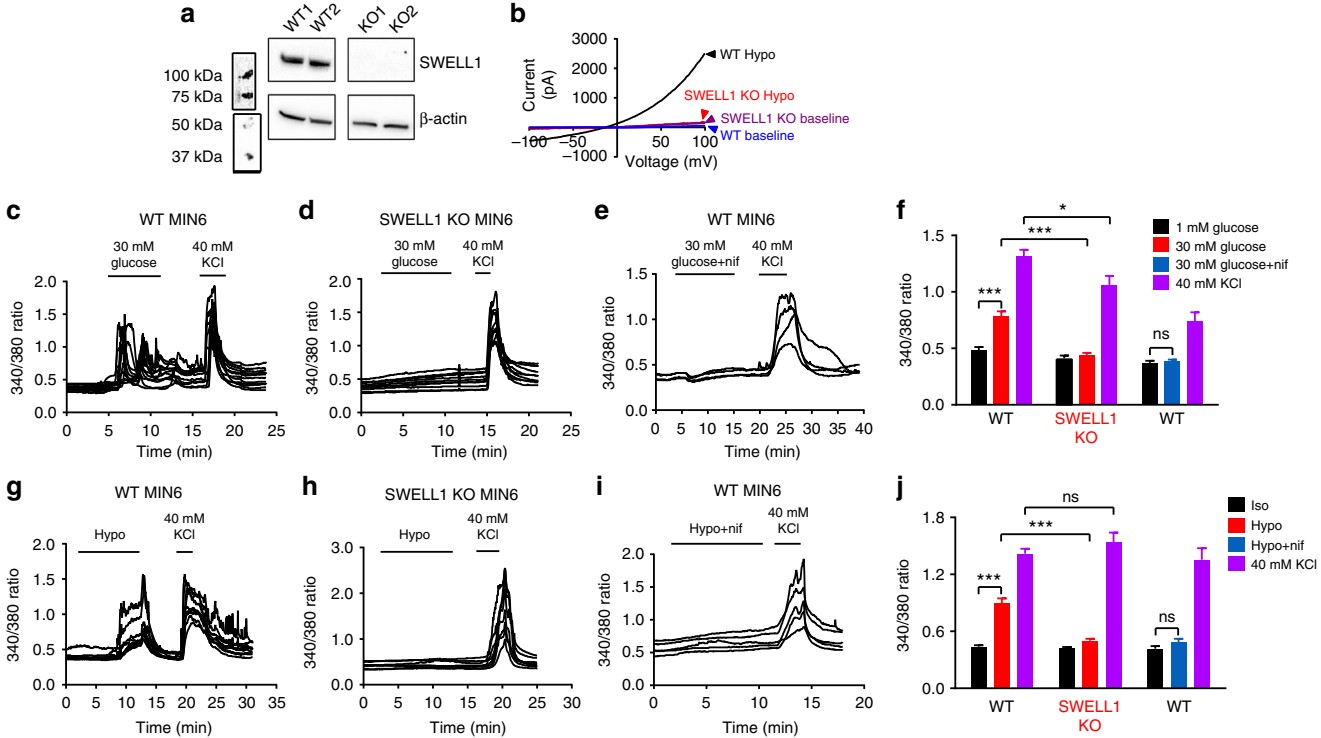

**Fig. 4** SWELL1-mediated $I_{Cl,SWELL}$ is required for both glucose- and hypotonic swelling-induced Ca$^{2+}$ signaling in MIN6 cells. **a** SWELL1 western blot in WT and CRISPR/Cas9-mediated *Swell1* KO MIN6 cell lines (Supplementary Fig. 6b for full blots). **b** Current–voltage plots of SWELL1-mediated current in response to hypotonic swelling in WT and *Swell1* KO MIN6 cells confirm complete ablation of hypotonic swelling-stimulated $I_{Cl,SWELL}$ in *Swell1* KO MIN6 cells. **c–e** Fura-2 Ca$^{2+}$ transients in **c** WT, **d** *Swell1* KO and **e** WT+nifedipine (10 μM) MIN6 cells in response to 30 mM glucose stimulation (basal 1 mM glucose). 40 mM KCl stimulation confirms cell viability and excitability. **f** Mean peak Fura-2 ratio in glucose-stimulated WT ($n = 34$), *Swell1* KO ($n = 36$), and WT+nifedipine ($n = 16$) MIN6 cells. **g–i** Fura-2 Ca$^{2+}$ transients in **g** WT, **h** *Swell1* KO, and **i** WT+nifedipine (10 μM) MIN6 cells in response to hypotonic swelling stimulation (210 mOsm/kg; isotonic 300 mOsm/kg). **j** Mean peak Fura-2 ratio in swell-stimulated WT ($n = 42$ cells), *Swell1* KO ($n = 26$ cells), and WT+nifedipine MIN6 cells. In the presence of VGCC blocker nifedipine (10 μM), neither glucose (**e**) nor hypotonic swelling (**i**) induced Ca$^{2+}$ transients in WT MIN6 cells. Data are shown as mean ± s.e.m. One-way ANOVA for in-group comparison, unpaired *t*-test for between-group comparison. *$p < 0.05$, ***$p < 0.001$. ns, not significant

from *Swell1*$^{fl/fl}$ mice followed by transduction with either Ad-RIP1-RFP (WT) or Ad-RIP1-Cre-P2A-RFP (*Swell1* KO; Fig. 7c). Similar to MIN6 cells, we note a significant reduction in GSIS (16.7 mM glucose) in *Swell1* KO compared to WT islets (Fig. 7d), despite relatively preserved L-arginine-stimulated insulin secretion (Fig. 7d), and similar total insulin content (Fig. 7e). Comparable to murine islets, SWELL1 KD human islets also exhibit a selective impairment in GSIS (Fig. 7f, g).

***Swell1* deletion impairs insulin release and systemic glycaemia in vivo**. We next generated tamoxifen(Tm)-inducible β-cell-targeted *Swell1* KO mice by crossing *Swell1*$^{fl/fl}$ mice with *Ins1*$^{CreERT2}$ mice (*Ins1*$^{CreERT2}$;*Swell1*$^{fl/fl}$, Fig. 8a)[39,40] to examine the requirement of β-cell SWELL1 for insulin secretion and regulation of systemic glycaemia in vivo. *Ins1*$^{CreERT2}$ mice have been previously characterized and validated to be indistinguishable from WT mice with respect to body weight and glucose homeostasis, while providing efficient, Tm-inducible, β-cell selective recombination[40]. Indeed, islets isolated from *Ins1*$^{CreERT2}$;*Rosa26-tdTomato*;*Swell1*$^{fl/fl}$ mice reveal efficient β-cell-targeted recombination after Tm-administration (40–80 mg/kg/day gavage × 5 days; Supplementary Fig. 5), similar to prior studies[40]. Moreover, we observe no evidence of β-cell loss upon *Swell1* deletion based on the number of *tdTomato*-positive β-cells present in the islet. We observe pancreas-restricted *Swell1* recombination in Tm-induced *Ins1*$^{CreERT2}$;*Swell1*$^{fl/fl}$ mice (+) by

PCR across *Swell1* Exon 3 (426 bp amplicon; Fig. 8b); and this recombination is not observed in vehicle-treated *Ins1*$^{CreERT2}$; *Swell1*$^{fl/fl}$ mice (−). Finally, we observe complete ablation of SWELL1-mediated $I_{Cl,SWELL}$ in 80% of β-cells isolated from Tm-induced *Ins1*$^{CreERT2}$;*Swell1*$^{fl/fl}$ (8/10 cells; Fig. 8c–d), while 100% of β-cells from vehicle-induced *Swell1*$^{fl/fl}$ (WT) exhibit robust hypotonically activated SWELL1-mediated currents (4/4 cells).

We next measured glucose tolerance in *Swell1*$^{fl/fl}$ and *Ins1*$^{CreERT2}$;*Swell1*$^{fl/fl}$ before and after Tm-induced β-cell selective *Swell1* deletion. Pre-Tm-induction, *Swell1*$^{fl/fl}$ (−) and *Ins1*$^{CreERT2}$; *Swell1*$^{fl/fl}$ (−) mice have identical fasting glucose levels (6 h fast) and glucose tolerance in response to a glucose load (Fig. 8e). However, post-Tm-treatment, *Ins1*$^{CreERT2}$;*Swell1*$^{fl/fl}$ (+) mice exhibit significantly impaired glucose tolerance compared to Tm-induced *Swell1*$^{fl/fl}$ (+) mice, predominantly soon after the glucose load (Fig. 8f). In a separate cohort of mice, we measured serum insulin in *Swell1*$^{fl/fl}$ and *Ins1*$^{CreERT2}$;*Swell1*$^{fl/fl}$ mice after Tm-induction (+). We find that basal (fasting) serum insulin values are similar between genotypes (Fig. 8g), however, with glucose stimulation insulin secretion is significantly impaired in Tm-induced *Ins1*$^{CreERT2}$;*Swell1*$^{fl/fl}$ (+) mice compared to Tm-induced *Swell1*$^{fl/fl}$ mice (+) (Fig. 8g, h), and this is also associated with impaired glucose tolerance (Fig. 8i). When these mice are placed on a high-fat diet (HFD; 60% fat) for 5–6 weeks to induce mild obesity, and a state of pre-diabetes, we find that Tm-induced *Ins1*$^{CreERT2}$;*Swell1*$^{fl/fl}$ (+) mice exhibit elevated fasting serum glucose (Fig. 8j) and exacerbated glucose intolerance (Fig. 8k) as

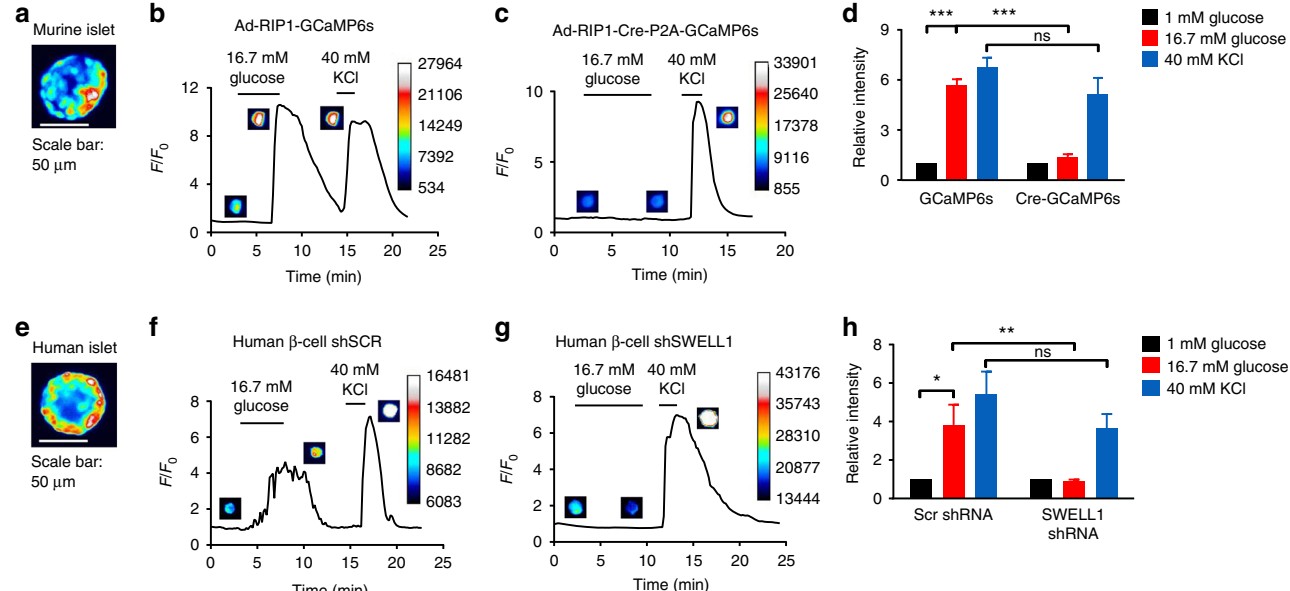

**Fig. 5** SWELL1-mediated $I_{Cl,SWELL}$ is required for glucose-stimulated $Ca^{2+}$ signaling in primary β-cells. **a** Ad-RIP1-GCaMP6s-transduced $Swell1^{fl/fl}$ murine islet. Scale bar represents 50 μm. **b–d** GCaMP6s $Ca^{2+}$ transients in WT (**b**, Ad-RIP1-GCaMP6s/$Swell1^{fl/fl}$) and SWELL1 KO (**c**, Ad-RIP1-Cre-P2A-GCaMP6s/$Swell1^{fl/fl}$) primary murine β-cells in response to 16.7 mM glucose stimulation (basal 1 mM glucose) and 40 mM KCl stimulation. Insets show β-cell fluorescence images at the indicated times in the experiment. **d** Mean peak values of GCaMP6s $Ca^{2+}$ transients from Ad-RIP1-GCaMP6s/$Swell1^{fl/fl}$ ($n = 14$ cells) and Ad-RIP1-Cre-P2A-GCaMP6s/$Swell1^{fl/fl}$ ($n = 10$ cells). **e** Ad-RIP1-GCaMP6s-transduced human islet. Scale bar represents 50 μm. **f–h** GCaMP6s $Ca^{2+}$ transients in human primary β-cells co-transduced with Ad-RIP1-GCaMP6s+Ad-shSCR-mCherry (**f**) and Ad-RIP1-GCaMP6s+Ad-sh$Swell1$-mCherry (**g**) in response to 16.7 mM glucose stimulation (basal 1 mM glucose) and 40 mM KCl stimulation. **h** Mean peak values of GCaMP6s $Ca^{2+}$ transients from shSCR ($n = 6$ cells) and sh$Swell1$ ($n = 8$ cells) transduced human β-cells. Data are shown as mean ± s.e.m. One-way ANOVA for in-group comparison, unpaired $t$-test for between-group comparison. $*p < 0.05$, $**p < 0.01$, $***p < 0.001$. ns, not significant

compared to Tm-induced $Swell1^{fl/fl}$ (+) mice, despite similar body weights (Fig. 8l). Collectively, these data reveal that SWELL1 is post developmentally required for intact glucose-stimulated β-cell membrane depolarization, $Ca^{2+}$-dependent insulin secretion and regulation of systemic glycaemia, particularly in the setting of mild obesity.

## Discussion

Ion channel regulation of β-cell excitability is critical for mediating β-cell $Ca^{2+}$ signaling, insulin secretion, and systemic glycaemia[2,41]. Therefore, identifying novel ion channels that control β-cell excitability will advance our understanding of β-cell physiology, and potentially open previously unexplored therapeutic avenues for the treatment of type 2 diabetes (T2D). VRAC or $I_{Cl,SWELL}$ is a swell-activated ionic current that has been studied for decades through electrophysiological recordings in numerous cell types[42–44], but only recently has it been discovered that SWELL1/LRRC8a, and associated LRRC8 isoforms b–e, form the $I_{Cl,SWELL}$ channel complex in common cell lines[21–23]. Accordingly, the physiological role of SWELL1-mediated $I_{Cl,SWELL}$ in primary cells remains largely unexplored. In the current study, we asked whether SWELL1 is required for VRAC/$I_{Cl,SWELL}$ described previously in the pancreatic β-cell[16,17,19] and whether the SWELL1 channel complex mediates a glucose- and swell-sensitive depolarizing current in β-cells, as initially proposed by Best et al.[14] Our data are consistent with a model in which SWELL1 is a required component of a swell-activated depolarizing $Cl^-$ channel that is present in MIN6 cells, in primary murine β-cells and, importantly, in primary human β-cells. SWELL1-mediated $I_{Cl,SWELL}$ activates in response to glucose-stimulated β-cell swelling and is required for normal membrane depolarization, VGCC activation, $Ca^{2+}$-mediated insulin vesicle fusion, and insulin secretion

(Fig. 9). In this model, the hyperpolarizing $K^+$ conductances[5–8,45,46] act as "brakes" on β-cell excitability and insulin secretion, while SWELL1-mediated $I_{Cl,SWELL}$ is the "accelerator"—promoting β-cell excitability in response to glucose-mediated β-cell swelling. We propose that it is the balance between $I_{Cl,SWELL}$ and $I_{KATP}$ that is critical in regulating β-cell membrane potential, as either robust $I_{Cl,SWELL}$ activation, or full $I_{KATP}$ inhibition, are alone capable of depolarizing the β-cell. However, partial $I_{KATP}$ inhibition (~30% K(ATP) blockade; 0.25 nM glibenclamide)[35] is incapable of stimulating β-cells in the absence of SWELL1-mediated $I_{Cl,SWELL}$. Our data are consistent with previous reports that postulate the existence of glucose sensitive anionic current(s), in addition to $I_{KATP}$ that contribute to glucose-stimulated β-cell depolarization[13,14,20,47–49]; especially since glucose-dependent $I_{KATP}$ inhibition occurs primarily at substimulatory glucose concentrations (0–5 mM) and is saturated in the stimulatory range between 5–20 mM glucose[48]. Thus, in our current model, SWELL1-mediated $I_{Cl,SWELL}$ and $I_{KATP}$ are both glucose sensitive ionic currents that antithetically regulate β-cell membrane potential.

The activation mechanism of SWELL1-mediated $I_{Cl,SWELL}$ is unique among other known ion channels expressed in the β-cell in that it is mediated by β-cell swelling[13,16,20], that under physiological conditions, is glucose stimulated[14,19], and dependent on glucose metabolism[19]. The mechanism of glucose-stimulated β-cell swelling is thought to occur via glucose break down into lactate[14,15] or other small metabolites, which increase intracellular osmotic pressure and draws in water[50], potentially via aquaporin 7 channels[51,52]. In this way, SWELL1-mediated $I_{Cl,SWELL}$ senses extracellular glucose indirectly, and is intrinsically coupled to glucose metabolism; just as $I_{KATP}$ is linked to glucose metabolism via glucose-dependent changes in ATP/ADP ratio. In addition, SWELL1-mediated $I_{Cl,SWELL}$ may also be

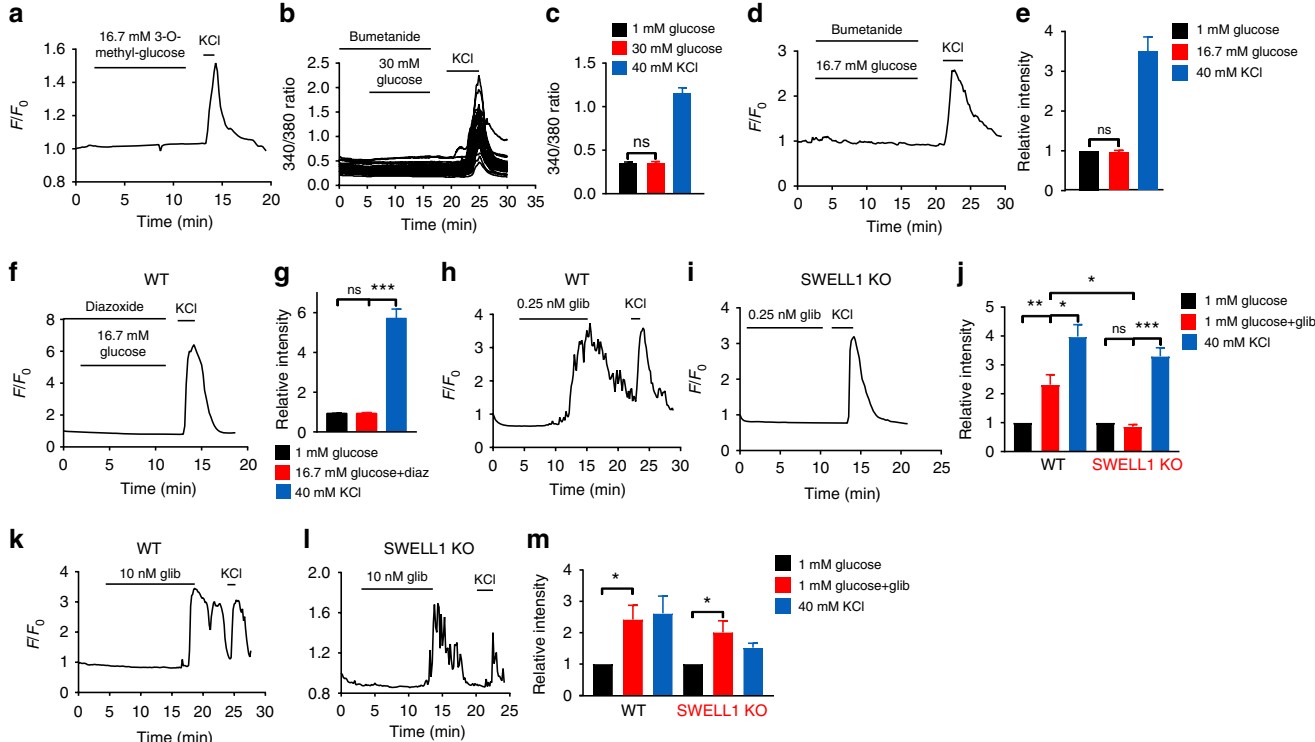

**Fig. 6** SWELL1-mediated $I_{Cl,SWELL}$ and $I_{KATP}$ coordinately regulate β-cell excitability. **a** GCaMP6s $Ca^{2+}$ transients in WT (Ad-RIP1-GCaMP6s/$Swell1^{fl/fl}$) primary murine β-cells in response to 16.7 mM 3-O-Methyl-glucose-stimulation (basal 1 mM glucose) and 40 mM KCl stimulation. 3-O-Methyl-glucose fail to induce $Ca^{2+}$ transients ($n = 15$ cells). **b**–**e** Glucose-induced $Ca^{2+}$ transients are inhibited in WT MIN6 cells (**b**, **c**) and in WT primary murine β-cells (**d**, **e**) by bumetanide (10 μM; pre-incubated in bumetanide for 2 h). **c** Mean peak Fura-2 ratio from bumetanide-treated WT MIN6 cells ($n = 46$ cells). **d** Mean peak values of GCaMP6s $Ca^{2+}$ transients from bumetanide-treated WT primary murine β-cells ($n = 7$ cells). **f**, **g** In the presence of $K_{ATP}$ channel opener, diazoxide (100 μM), glucose-induced $Ca^{2+}$ transients are abolished in WT primary murine β-cells. **g** Mean peak values of GCaMP6s $Ca^{2+}$ transients from WT primary murine β-cells treated with diazoxide ($n = 17$ cells). **h**–**j** Low-dose glibenclamide (0.25 nM, 0 mM glucose), which is predicted to block a fraction of $K_{ATP}$ channels, reproducibly induced $Ca^{2+}$ transients in WT (**h**), but not $Swell1$ KO (**i**) primary murine β-cells. **j** Mean peak values of GCaMP6s $Ca^{2+}$ transients stimulated by 0.25 nM glibenclamide in WT β-cells ($n = 19$ cells) and $Swell1$ KO β-cells ($n = 8$ cells), and KCl-positive controls. **k**–**m** High-dose glibenclamide (10 nM), which has a maximal inhibitory effect on $K_{ATP}$ channels, activated $Ca^{2+}$ transients in both WT (**k**) and $Swell1$ KO β-cells (**l**). **m** Mean peak values of GCaMP6s $Ca^{2+}$ transients stimulated by 10 nM glibenclamide in WT β-cells ($n = 4$ cells) and $Swell1$ KO β-cells ($n = 4$ cells). Data are shown as mean ± s.e.m. One-way ANOVA for in-group comparison, unpaired $t$-test for between-group comparison. *$p < 0.05$, **$p < 0.01$, ***$p < 0.001$. ns, not significant

modulated by intracellular ATP, in addition to β-cell swelling[53], providing another putative mechanism for metabolic regulation of $I_{Cl,SWELL}$.

Notably, SWELL1-deficient β-cells exhibit preserved glucose-stimulated swelling, suggesting that glucose metabolism is intact in these cells[15,19]. Hence, it is unlikely that impaired β-cell excitability in SWELL1-deficient β-cells arises from defective glucose metabolism. Indeed, β-cell-targeted disruption of glucose metabolism through β-cell selective deletion of glucokinase induces a very severe diabetic phenotype with early neonatal lethality[54], in contrast to milder, glucose intolerance observed with tamoxifen-induced β-cell-targeted $Swell1$ deletion.

The biophysical properties of the SWELL1 channel complex mediating $I_{Cl,SWELL}$ is intriguing with respect to β-cell electrophysiology and regulation of excitation–secretion coupling. The property of outward rectification provides greater β-cell membrane depolarization at hyperpolarized potentials (i.e., larger inward currents) to initiate β-cell excitation, but then electrochemically "shuts-off" at more depolarized potentials as the membrane potential approaches the equilibrium potential of $Cl^-$ ($E_{Cl^-}$ ~−15 mV[29,32]). As the membrane potential exceeds $E_{Cl^-}$, SWELL1-mediated $I_{Cl,SWELL}$ provides a hyperpolarizing current to stabilize the β-cell around $E_{Cl^-}$, thereby maximizing $Ca^{2+}$ influx via voltage-gated $Ca^{2+}$ channels, as hypothesized by

Eberhardson et al.[28], and also allows for β-cell repolarization, and termination of insulin secretion upon subsequent activation of delayed-rectifier potassium channels[6]. Interestingly, the RVD mediated by $I_{Cl,SWELL}$/VRAC activation promotes β-cell contraction (Fig. 2a and Supplementary Fig. 2a)[19], thereby removing the swell stimulus for $I_{Cl,SWELL}$ activation. This results in SWELL1 channel complex closure upon return to basal glucose, which also permits β-cell repolarization.

In addition to SWELL1-mediated $I_{Cl,SWELL}$, there are certainly other depolarizing currents that contribute to β-cell excitability, since full suppression of $I_{KATP}$ is capable of stimulating $Swell1$ KO β-cells. These include TRP channels, including TRPM3[12], TRPM4[55], and TRPM5[9–11]; though not all of these TRP channels are necessarily expressed in human β-cells[56]. Cystic fibrosis transmembrane conductance regulator (CFTR) has been implicated as a $Cl^-$ conductance important for glucose-stimulated β-cell excitability and secretion[57] as has ANO1/TMEM16A[58]; however, the presence of CFTR in the β-cell is controversial[59]. Finally, ClC-3, a chloride channel localized to insulin granules, has been proposed to regulate insulin processing and secretion via acidification mechanisms[60]. Given the significant impact of SWELL1 on β-cell excitability, insulin secretion and systemic glycaemia, it is possible that SWELL1 may have other, as yet unrealized, ion channel regulatory functions in β-cell biology.

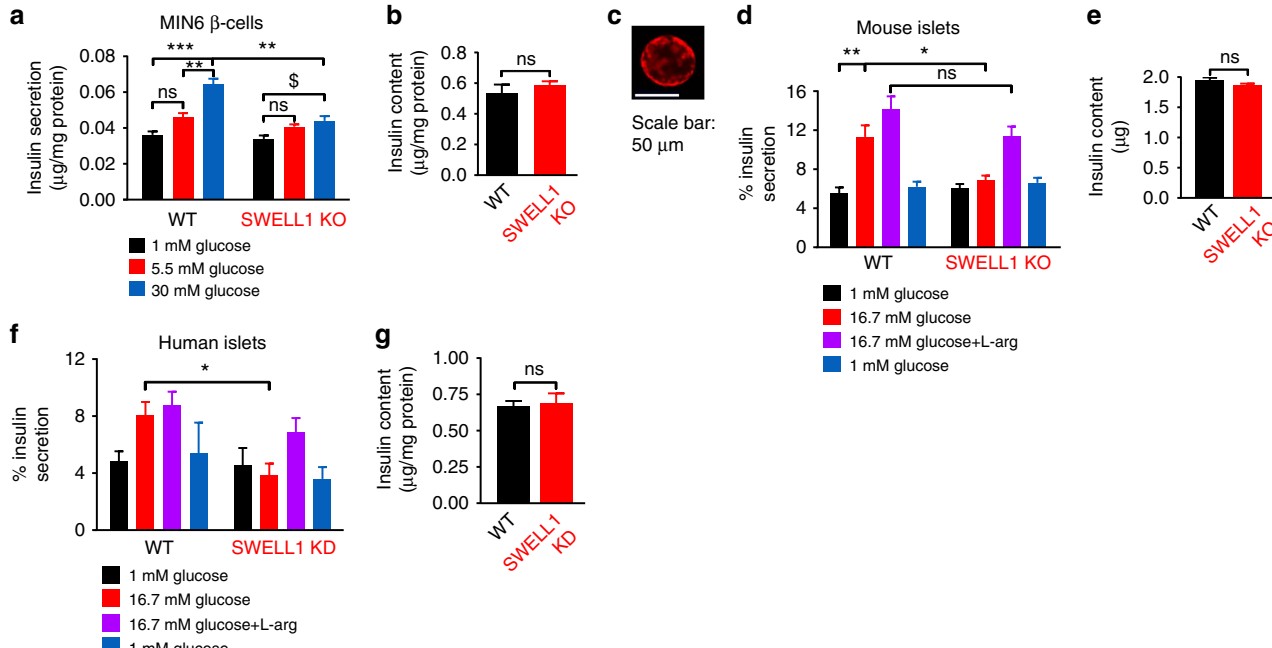

**Fig. 7** β-cell SWELL1 is required for glucose-stimulated insulin secretion in vitro. **a** Glucose-stimulated insulin secretion (GSIS) in WT and *Swell1* KO MIN6 cells in response to 1, 5.5, and 30 mM glucose ($n = 3$, each). **b** Total insulin content normalized to total MIN6 cell protein content ($n = 3$, each). **c** Representative image of Ad-RIP1-Cre-RFP-transduced mouse islet. Scale bar represents 50 μm. **d** GSIS in WT (Ad-RIP1-RFP/*Swell1*^fl/fl^) and β-cell-targeted SWELL1 KO (Ad-RIP1-Cre-RFP/*Swell1*^fl/fl^) islets. Insulin secretion is expressed as percentage of the total insulin content ($n = 4$, each). **e** Total insulin content of WT and β-cell-targeted *Swell1* KO islets ($n = 4$, each). **f** GSIS in human WT and SWELL1 KD islets. Insulin secretion is expressed as percentage of the total insulin content ($n = 4$, each). **g** Total insulin content of WT and SWELL1 KD islets ($n = 4$, each). Data are shown as mean ± s.e.m. One-way ANOVA for in-group comparison, unpaired *t*-test for between-group comparison. *$p < 0.05$, **$p < 0.01$, ***$p < 0.001$. ns, not significant

Our main findings are that SWELL1-mediated $I_{Cl,SWELL}$ is a glucose/swell-activated depolarizing ionic current that is required for GSIS both in vitro and in vivo. Under non-pathological conditions, in lean mice, SWELL1-mediated $I_{Cl,SWELL}$ is primarily required for maintaining glycaemia in response to a glucose challenge—inducing a state of β-cell-dependent pre-diabetes. However, with super-imposed mild insulin resistance arising from a short course of high-fat diet (60% fat for 6 weeks), SWELL1-mediated $I_{Cl,SWELL}$ becomes important for maintaining systemic fasting glycaemia, suggesting that SWELL1 may be primarily required to sustain β-cell insulin secretion in hyperglycemic states, as occur in the setting of T2D. Indeed, T2D islets exhibit reduced GSIS by up to 60%[1,2] and this remains true even when expressed relative to total insulin content[3]. Moreover, this impairment in GSIS from T2D β-cells is largely recoverable by sulfonylurea, L-arginine or KCl treatment in vitro[61]; similar to our findings in *Swell1* KO β-cells/islets. Taken together, these data raise the intriguing possibility that impaired β-cell function in T2D may arise from deficient SWELL1-mediated $I_{Cl,SWELL}$ in β-cells. We are currently testing this hypothesis.

In summary, our data suggest that β-cell SWELL1 acts as a glucose sensor by coupling β-cell swelling to β-cell depolarization —a form of swell activation or swell secretion coupling—to potentiate GSIS. In the broader context of our findings on SWELL1 signaling in the adipocyte[25], these data suggest that SWELL1 coordinately regulates both insulin secretion and insulin sensitivity[25] in response to a nutrient load. These findings highlight the importance of SWELL1 in the regulation of systemic glucose metabolism, and organismal energy homeostasis, particularly in the setting of obesity.

## Methods

**Animals**. The Institutional Animal Care and Use Committee of the University of Iowa approved all experimental procedures involving animals. All the mice were housed in a temperature, humidity, and light-controlled room and allowed free access to water and food. Both male and female (49% males; 51% females) *Swell1*^fl/fl^, *Ins1*^CreERT2^;*Swell1*^fl/fl^, and *Ins1*^CreERT2^;*Rosa26-tdTomato*;*Swell1*^fl/fl^ mice, ages 6–20 weeks were used for in vitro and for in vivo experiments. In a subset of experiments, 8–10-week-old *Swell1*^fl/fl^ and *Ins1*^CreERT2^;*Swell1*^fl/fl^ mice were switched to a HFD (HFD rodent diet with 60 kcal% fat, Research Diets, Inc., D12492) for at least 5 weeks.

**Antibodies and ion channel modulators**. Rabbit polyclonal anti-SWELL1 antibody was generated against the epitope QRTKSRIEQGIVDRSE (Pacific Antibodies). Rabbit monoclonal anti-β-actin antibody was purchased from Cell Signaling (8457). Primary antibodies were used at 1:1000 dilution. 4-[(2-Butyl-6,7-dichloro-2-cyclopentyl-2,3-dihydro-1-oxo-1*H*-inden-5-yl)oxy]butanoic acid (DCPIB) was purchased from TOCRIS (#2A/157889); nifedipine from Sigma-Aldrich (N7634); bumetanide from Sigma-Aldrich (B3023); glibenclamide from Sigma-Aldrich (G0639); diazoxide from Sigma-Aldrich (D9035); and 3-O-Methyl-glucose from Sigma-Aldrich (M4879). Stock solutions of drugs were made in DMSO (Sigma-Aldrich, D8418) and diluted to desired concentrations immediately prior to use.

**Adenoviruses and adeno-associated virus preparation**. Human adenoviruses type 5 with hLRRC8A-shRNA (Ad5-mCherry-U6-hLRRC8A-shRNA, $2.2 \times 10^{10}$ PFU/ml), a scrambled non-targeting control (Ad5-U6-scramble-mCherry, $1 \times 10^{10}$ PFU/ml), β-cell-targeted GCaMP6s (Ad5-RIP1-GCaMP6s, $4.9 \times 10^{10}$ PFU/ml), β-cell-targeted GCaMP6s-2A-iCre (Ad5-RIP1-GCaMP6s-2A-iCre, $5.8 \times 10^{10}$ PFU/ml), β-cell-targeted GFP (Ad5-RIP2-GFP, $4.1 \times 10^{10}$ PFU/ml), β-cell-targeted RFP (Ad5-RIP1-RFP, $1.9 \times 10^{10}$ PFU/ml), and β-cell-targeted RFP-2A-iCre (Ad5-RIP1-RFP-2A-Cre, $6.4 \times 10^{10}$ PFU/ml) were obtained from Vector Biolabs. Human adenovirus type 5 with mCherry (Ad5-CMV-mCherry; $1 \times 10^{10}$ PFU/ml) and Cre-mCherry (Ad5-CMV-mCherry, $1 \times 10^{10}$ PFU/ml) were obtained from The University of Iowa Viral Vector Core.

**Cell culture**. No cell lines used in this study were found in the database of commonly misidentified cell lines that is maintained by ICLAC and NCBI biosample. MIN6 cells were kindly provided by Dr Robert Tsushima (York University, Canada) and were cultured in Dulbecco's Modified Eagle Medium (DMEM) containing 4.5 g/l glucose, 10% FBS, 1% L-glutamine and 100 IU penicillin and 100 μg/ml streptomycin. Cells were grown in culture dishes at 37 °C, 5% $CO_2$ and 95% air. Cell lines were tested negative for mycoplasma. For electrophysiological recordings and intracellular calcium imaging, cells were seeded onto collagen-

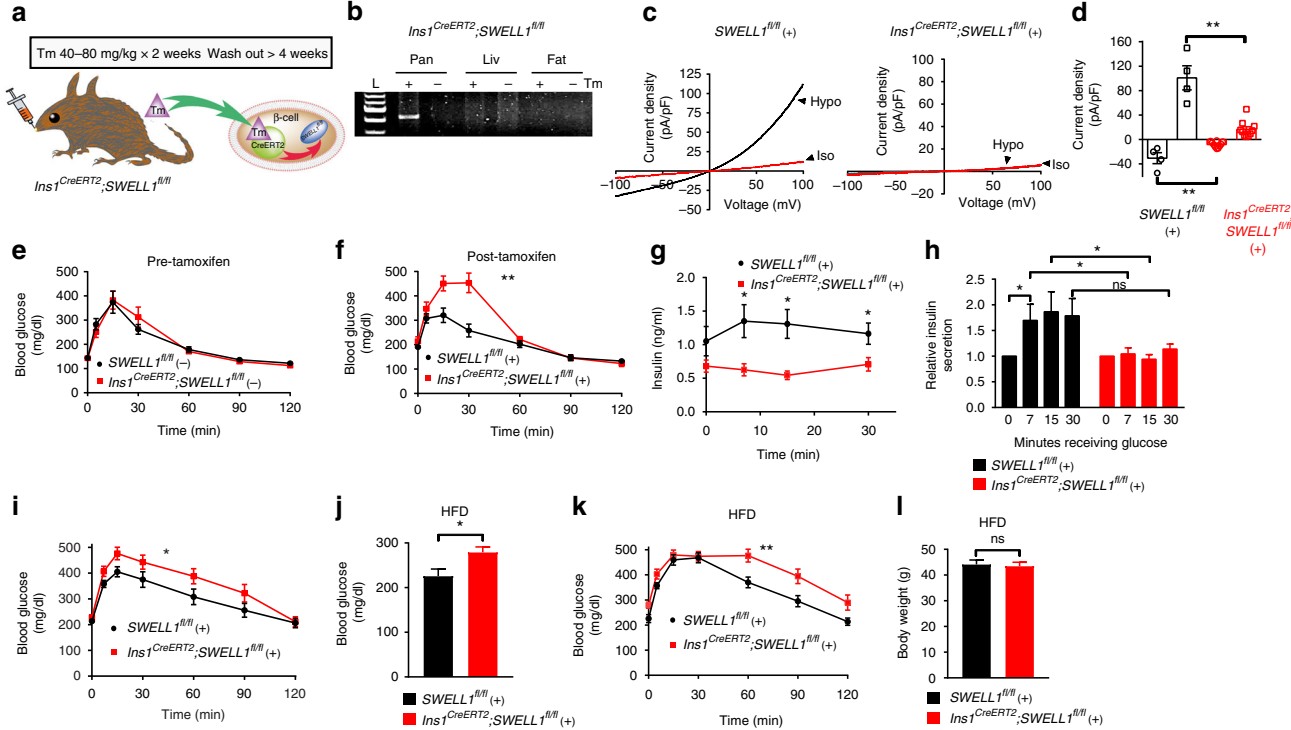

**Fig. 8** Tamoxifen-inducible, β-cell-targeted *Swell1* deletion impairs glucose-stimulated insulin secretion and systemic glycaemia. **a** Tamoxifen(Tm)-inducible β-cell-targeted *Swell1* inactivation using *Ins1*$^{CreERT2}$ mouse crossed with *Swell1*$^{fl/fl}$ mice (*Ins1*$^{CreERT2}$; *Swell1*$^{fl/fl}$). **b** PCR across *Swell1* Exon 3 in genomic DNA extracted from pancreas, liver, and adipose tissues of *Ins1*$^{CreERT2}$;*Swell1*$^{fl/fl}$ mice treated with (+) or without (−) tamoxifen. *Cre*-mediated *Swell1* recombination results in a 426 bp amplicon observed only in DNA extracted from the pancreas of Tm(+) treated *Ins1*$^{CreERT2}$;*Swell1*$^{fl/fl}$ mice. **c** Current–voltage relationship of hypotonic swell-activated $I_{Cl,SWELL}$ in mouse primary β-cells isolated from *Swell1*$^{fl/fl}$ treated with Tm (+, left) and *Ins1*$^{CreERT2}$; *Swell1*$^{fl/fl}$ treated with Tm (+, right). **d** Mean inward and outward current densities at +100 and −100 mV from Tm-treated *Swell1*$^{fl/fl}$ (+; n = 4 cells) and *Ins1*$^{CreERT2}$;*Swell1*$^{fl/fl}$ (+; n = 10 cells) mice. **e**, **f** Glucose tolerance tests (GTT) (2 g/kg glucose i.p.) in *Ins1*$^{CreERT2}$; *Swell1*$^{fl/fl}$ mice pre (**e**, −) and post (**f**, +) Tm treatment, respectively (*Swell1*$^{fl/fl}$, n = 7 mice; *Ins1*$^{CreERT2}$;*Swell1*$^{fl/fl}$, n = 6 mice) raised on a regular-chow diet. **g**, **h** Glucose-stimulated insulin secretion in Tm-treated *Swell1*$^{fl/fl}$ (+) and *Ins1*$^{CreERT2}$;*Swell1*$^{fl/fl}$ (+) mice in response to 2 g/kg glucose in the first 30 min (*Swell1*$^{fl/fl}$, n = 20 mice; *Ins1*$^{CreERT2}$;*Swell1*$^{fl/fl}$, n = 22 mice). **i** GTT (2 g/kg glucose i.p.) from *Swell1*$^{fl/fl}$ (+) and *Ins1*$^{CreERT2}$;*Swell1*$^{fl/fl}$ (+) mice (n = 15, in each group) on a regular-chow diet. **j** Fasting serum blood glucose, **k** GTT (1 g/kg i.p.) and **l** body mass of the same cohort of mice in **i** after 5–6 weeks on HFD (60% fat, n = 15, in each group). Data are shown as mean ± s.e.m. In **d**, **g**, **j**, **l**, unpaired *t*-test are performed. In **h**, one-way ANOVA for in-group comparison, unpaired *t*-test for between-group comparison. In **e**, **f**, **i**, **k**, two-way ANOVA are performed. *p < 0.05; **p < 0.01; ***p < 0.001. ns, not significant

coated (Millipore, USA) glass cover slips and used for experiments after 24–48 h at 30–40% confluency.

**CRISPR/Cas9-mediated *Swell1* knockout in MIN6 cells**. A CRISPR/Cas9-based tool was utilized to knockout *Swell1* gene in MIN6 cells, where the target guide sequences designed using web-based CRISPR design tool (http://crispr.mit.edu/) were cloned into a bicistronic vector-expressing cas9 (pSpCas9(BB)-2A-Puro)[62]. Either 1A/2B (KO1) or 2B/3C (KO2) plasmids in combination were transfected into MIN6 cells using LipofectAMINE 2000 as per the manufacturer's instructions (Supplementary Table 1). After a 48 h period, selection medium containing pur-omycin (1 µg/ml) was added to the cells and maintained for 5 days. Individual clones from the enriched pool were further isolated by dilution method[62]. *Swell1* gene deletion was confirmed by PCR amplification spanning the double guide targeted region (Supplementary Fig. 3), followed by SWELL1 western blot (Fig. 4a and Supplementary Fig. 6b).

**Human islets**. Human islets were obtained through the Integrated Islet Distribution Program (IIDP, shared with Dr John Engelhardt) or Prodo Laboratories (shared with Dr Yumi Imai). Ethical approval was not required for the in vitro studies using human islets obtained from IIDP and Prodo Laboratories. Patients were anonymous to the research team except for information of gender, age, and BMI. Islets were cultured in RPMI 1640 (2% FBS) and transferred to the laboratory within 24 h.

**Murine islet isolation**. *Swell1*$^{fl/fl}$ mice (8–14 weeks old) were killed by Avertin (0.0125 g/ml, dissolved in $H_2O$) injection (20 µl/g, i.p.) followed by cervical dis-location according to the approved procedures. The pancreas was perfused via the common bile duct with 2–3 ml HBSS containing type V collagenase (0.8 mg/ml), and then removed and digested at 37 °C for 10 min. Islets were freed by gentle

agitation, washed in RPMI containing 1% FBS and purified on Histopaque 1077 and 1119 gradients. Islets were then transferred to 60 mm Petri dish with culture medium for short-term culture (<24 h). For GSIS experiments, islets were sorted for equal size and cultured in 24-well plates. For isolation of primary β-cells, isolated islets were further incubated in trypsin for 5 min, dispersed into single cells, and then transferred to matrigel-coated cover slips for patch clamp and calcium fluorescence imaging.

**Adenoviral transduction of murine and human islets**. Human and murine islets were cultured in RPMI media with 2% FBS overnight. The next day, adenovirus was added into the islet containing media (final concentration of $5 \times 10^7$ PFU/ml) and islets incubated for 24 h. The islets were then washed with PBS three times and cultured in RPMI medium with 10% FBS for 4–5 days before performing further experiments. Transduction efficiency was assessed by fluorescence microscopy.

**Western blotting studies**. Western blotting studies were carried out with lysates prepared from MIN6 cells by using standard techniques[63]. The primary antibodies used are listed above in antibodies and ion channel modulators. Cultured wild-type (WT) and the *Swell1* knockout (KO) MIN6 cells were washed twice with ice-cold phosphate buffer saline (PBS) and lysed using RIPA buffer (NaCl: 150 mM, HEPES: 20 mM, NP-40: 1%, EDTA: 5 mM, pH 7.4) containing protease and phosphatase inhibitors. The lysate was further clarified by sonication for about 10–20 s of two–three cycles. The supernatant was collected from the whole lysate centrifuged at 14,000 rpm for 15 min at 4 °C. Protein concentration was estimated by DC protein assay kit as per the manufacturer's (Bio-Rad) instructions. For protein detection using western blot method, 20–50 µg of protein was boiled in SDS loading buffer and separated using 4–15% SDS-PAGE and further transferred onto a PVDF membrane. The membrane were either blocked with 5% BSA or 5% milk containing TBST buffer (Tris: 0.2 M, NaCl: 1.37 M, Tween-20: 0.2%, pH 7.4) for 1 h

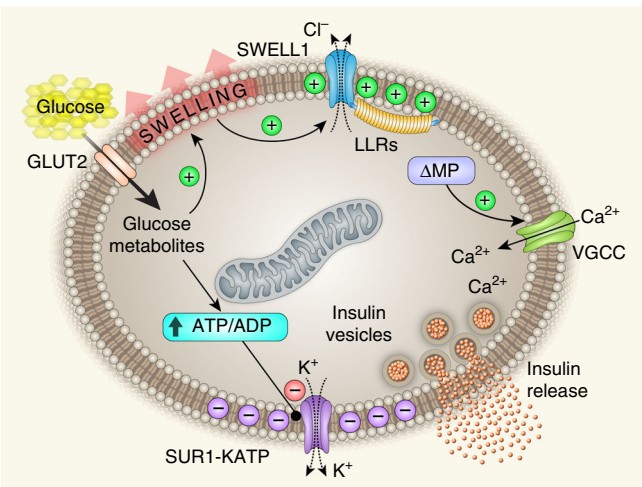

**Fig. 9** Working model: SWELL1 is a virtual glucose sensor regulating β-cell excitability and systemic glycaemia. GLUT2, glucose transporter; ΔMP, membrane depolarization; VGCC, voltage-gated calcium channel

at room temperature and probed with appropriate primary antibodies by incubating them at 4 °C overnight. Membrane was further incubated with appropriate secondary antibody (Bio-Rad, goat-anti-rabbit #170-6515) for 1 h at room temperature. The membranes were visualized by chemiluminescence using the ChemiDoc XRS+imaging system (Bio-Rad).

**Calcium imaging**. MIN6 cells were loaded with Fura-2-AM (10 μM) in DMEM at 37 °C for 20 min and then incubated with the basal isotonic Krebs-Ringer Bicarbonate HEPES (KRBH) buffer for 30 min. For calcium imaging experiments with hypotonic swelling, isotonic KRBH buffer consisted of the following (in mM): 90 NaCl, 5 NaHCO$_3$, 4.8 KCl, 1.2 KH$_2$PO$_4$, 2.5 CaCl$_2$, 2.4 MgSO$_4$, 10 HEPES, 90 mannitol, 0.1% w/v bovine serum albumin, pH 7.4 with NaOH (300 mOsm/kg); hypotonic KRBH buffer was (in mM): 90 NaCl, 5 NaHCO$_3$, 4.8 KCl, 1.2 KH$_2$PO$_4$, 2.5 CaCl$_2$, 2.4 MgSO$_4$, 10 HEPES, 0.1% w/v bovine serum albumin, pH 7.4 with NaOH (220 mOsm/kg). For glucose-stimulated intracellular calcium signaling, the basal KRBH solution was (in mM): 129 NaCl, 5 NaHCO$_3$, 4.8 KCl, 1.2 KH$_2$PO$_4$, 2.5 CaCl$_2$, 2.4 MgSO$_4$, 10 HEPES, 1 glucose, 29 mannitol, 0.1% w/v bovine serum albumin, pH 7.4 with NaOH (300 mOsm/kg). In MIN6 cells, we stimulated from 1 mM glucose basal solution to 30 mM glucose, and with primary β-cells from 1 to 16.7 mM glucose. Osmolarity was matched by adjusting with mannitol. For glibenclamide-stimulated intracellular calcium signaling, the basal KRBH solution was (in mM): 129 NaCl, 5 NaHCO$_3$, 4.8 KCl, 1.2 KH$_2$PO$_4$, 2.5 CaCl$_2$, 2.4 MgSO$_4$, 10 HEPES, 30 mannitol, 0.1% w/v bovine serum albumin, pH 7.4 with NaOH (300 mOsm/kg). All imaging was performed at 35–37 °C. MIN6 cells were imaged every 3 s using a 20x/0.75 NA objective (Olympus, Japan) on an Olympus IX73 microscope (Olympus, Japan), alternatively excited by 340 and 387 nm light using a DG-4 xenon-arc lamp (Sutter Instruments, USA). Emission signals were recorded at 510 nm and images obtained using a CMOS charge-coupled device (CCD) camera (Orca flash 4.0+, Hamamatsu, Japan). Intracellular calcium is represented as the change in the ratio of 340/387 fluorescent signal intensity. For primary mouse and human β-cells, dissociated β-cells from Ad-RIP1-GCaMP6s (Swell1$^{fl/fl}$ mouse/human) or Ad-RIP1-GCaMP6s-2A-iCre (Swell1$^{fl/fl}$ mouse) transduced islets were imaged as above but were imaged every 10 s via 485 nm excitation and 520 nm emission filters and relative changes in calcium concentration expressed as $F/F_o$.

**Cell volume measurements**. In experiments to study the effects of glucose-induced β-cell swelling, dissociated WT and SWELL1-deficient primary mouse and human β-cells identified from Ad-RIP1-GCaMP6s or Ad-RIP1-GCaMP6s-2A-iCre (Swell1$^{fl/fl}$ β-cells) transduced islets were imaged in bright field. Acquisitions were performed on an Olympus IX73 microscope at 37 °C, using a 40x/0.60 NA objective, connected to a CMOS CCD camera. Images were acquired every 10 s and captured using MetaMorph (Molecular Devices) software. β-cells were first perfused with basal KRBH solution consisting of (in mM): 129 NaCl, 5 NaHCO$_3$, 4.8 KCl, 1.2 KH2PO4, 2.5 CaCl$_2$, 2.4 MgSO$_4$, 10 HEPES, 1 glucose, 29 mannitol, 0.1% w/v bovine serum albumin, pH 7.4 with NaOH (300 mOsm/kg). Glucose concentration was increased from 1 to 16.7 mM. Osmolarity was matched by adjusting with mannitol. Quantification of β-cell cross-sectional area was performed from bright-field images using ImageJ (NIH) using an automated tool and expressed as total cross-sectional area ($A$) over initial area ($A_o$).

**Electrophysiology**. Patch clamp recordings were performed using either an Axopatch 200B amplifier or a MultiClamp 700B amplifier paired to a Digidata

1550 digitizer—both using pClamp 10.4 software. For hypotonic swelling, extracellular solution contained (in mM): 90 NaCl, 2 CsCl, 1 MgCl$_2$, 1 CaCl$_2$, 10 HEPES, 10 mannitol, pH 7.4 with NaOH (210 mOsm/kg). The isotonic extracellular solution consisted of the same composition above but with 110 mM instead of 10 mM mannitol (300 mOsm/kg). The intracellular solution contained (in mM): 120 L-aspartic acid, 20 CsCl, 1 MgCl$_2$, 5 EGTA, 10 HEPES, 5 MgATP, 120 CsOH, 0.1 GTP, pH 7.2 with CsOH. Swell-activated currents were elicited by perfusing cells with hypotonic solution (220 mOsm/kg) at room temperature in whole-cell configuration

For recording glucose-stimulated $I_{Cl,SWELL}$ currents, the basal extracellular solution contained (in mM): 90 NaCl, 2 CsCl, 1 MgCl$_2$, 1 CaCl$_2$, 109 mannitol, 1 glucose, ±0.01 nifedipine, pH 7.4 with NaOH (300 mOsm/kg). For glucose stimulation, glucose was increased from 1 to 16.7 mM (primary mouse and human β-cells). Osmolarity was matched with mannitol. The patch electrodes were prepared from borosilicate glass capillaries (WPI) and had a resistance of 3.0–5.0 MΩ when filled with pipette solution. For perforated patch recordings, the intracellular solution was as above but without ATP and GTP, and contained 240 μg/ml amphotericin B (Sigma-Aldrich, A9528). The holding potential was 0 mV. Voltage ramps from +100 to −100 mV (at 0.4 mV/ms) were applied every 4 s. Sampling interval was 100 μs and filtered at 10 KHz. Cells with a membrane resistance below GΩ or access resistance above 25 MΩ were discarded.

For measuring β-cell membrane potential upon hypotonic swelling stimulation, we used the whole-cell patch clamp configuration in current-clamp mode. Cells were treated with either isotonic or hypotonic KRBH buffer. Isotonic KRBH buffer consisted of the following (in mM): 90 NaCl, 5 NaHCO$_3$, 4.8 KCl, 1.2 KH$_2$PO$_4$, 2.5 CaCl$_2$, 2.4 MgSO$_4$, 10 HEPES, 90 mannitol, 0.1% w/v bovine serum albumin, pH 7.4 with NaOH (300 mOsm/kg). Hypotonic KRBH buffer contained (in mM): 90 NaCl, 5 NaHCO$_3$, 4.8 KCl, 1.2 KH$_2$PO$_4$, 2.5 CaCl$_2$, 2.4 MgSO$_4$, 10 HEPES, 0.1% w/v bovine serum albumin, pH 7.4 with NaOH (220 mOsm/kg). Pipettes were filled with (in mM): 100 L-aspartic acid, 40 KCl, 1 MgCl$_2$, 10 EGTA, and 10 HEPES (pH 7.25, adjusted with KOH), 295 mOsm/kg. Experiments were performed at room temperature.

For measuring glucose-stimulated changes in β-cell membrane potential, we used the perforated patch configuration in current-clamp mode. Cells were treated with 1 or 16.7 mM glucose dissolved in Krebs-Ringer-buffer (KRB) of the following composition (in mM): 129 NaCl, 5 NaHCO$_3$, 4.8 KCl, 1.2 KH$_2$PO$_4$, 2.5 CaCl$_2$, 2.4 MgSO$_4$, 10 HEPES, and 0.1 % w/v BSA (pH 7.4, adjusted with NaOH), osmolarity was 300 mOsm/kg matched with mannitol. The pipettes were filled with solution (in mM): 140 KCl, 1 MgCl$_2$, 10 EGTA, and 10 HEPES (pH 7.25, adjusted with KOH), 295 mOsm/kg. All glucose stimulation experiments were carried out at 35–37 °C.

**Insulin secretion and protein content assays**. GSIS from MIN6 cells and mouse islets was determined using a static incubation protocol. For MIN6 cells, cells were cultured in 6-well plates until ~50% confluent, and the media was changed every 24–48 h. On the day of experiment, cell culture media was removed, and the cells were washed twice with PBS and then pre-incubated with glucose-free KRBH buffer supplemented with 0.1% bovine serum albumin for 1 h at 37 °C and in 5% CO$_2$. MIN6 cells were then pre-incubated in KRBH buffer (1 mM glucose, 300 mOsm/kg) for 2 h at 37 °C and in 5% CO$_2$. After removal of pre-incubation solution, MIN6 cells were incubated in KRBH containing 1 mM, 5.5 mM, or 30 mM glucose for 1 h at 37 °C and in 5% CO$_2$. For all experiments, incubation media was collected, and the amount of secreted insulin was determined using ELISA (Mercodia, Sweden). After completion of the incubations, the MIN6 cells were lysed by addition of RIPA buffer, and the protein content was determined using the DC protein assay (Bio-Rad, USA).

Insulin secretion from isolated islets was determined using a similar protocol. Briefly, isolated islets of similar size were handpicked and pre-incubated with KRBH (1 mM glucose, 300 mOsm/kg) for 1 h in cell culture inserts within a 24-well plate (Falcon, USA) at 37 °C and in 5% CO$_2$. After the pre-incubation period, pre-incubation solutions were removed, inserts were then moved to fresh wells and KRBH containing glucose (1, 5.5, or 16.7 mM) or glucose (16.7 mM) plus L-arginine (Sigma-Aldrich, A8094) was added into the inserts. The islets were then incubated for 1 h at 37 °C and in 5% CO$_2$. After incubation, the medium was collected and the amount of secreted insulin determined using ELISA as described above. At the completion of the experiment, islets were lysed by addition of RIPA buffer and the amount of insulin detected by ELISA. MIN6 cell and mouse islet data sets were obtained from four independent experiments.

**Generation of CRISPR/Cas9-mediated Swell1 floxed (Swell1$^{fl/fl}$) mice**. Swell1$^{fl/fl}$ mice were generated as previously described[63]. Briefly, Swell1 intronic sequences were obtained from Ensembl Transcript ID ENSMUST00000139454. All CRISPR/Cas9 sites were identified using ZiFit Targeter Version 4.2 (http://zifit.partners.org/ZiFiT/). Pairs of oligonucleotides corresponding to the chosen CRISPR/Cas9 target sites were designed, synthesized, annealed, and cloned into the pX330-U6-Chimeric_BB-CBh-hSpCas9 construct (Addgene plasmid # 42230), following the protocol detailed in Cong et al.[64]. CRISPR-Cas9 reagents and ssODNs were injected into the pronuclei of F1 mixed C57/129 mouse strain embryos at an injection solution concentration of 5 ng/μl and 75–100 ng/μl, respectively. Correctly targeted mice were screened by PCR across the predicted loxP insertion sites

on either side of Exon 3. These mice were then backcrossed >8 generations into a C57B6 background.

**Generation of tamoxifen-inducible β-cell-targeted *Swell1* KO mice**. The generation of homozygous floxed *Swell1* mice (*Swell1$^{fl/fl}$* mice) in which exon 3 is flanked by loxP sites is described above. Tamoxifen-inducible β-cell-targeted *Swell1* KO were generated by crossing *Swell1$^{fl/fl}$* mice with *Ins1$^{CreERT2}$* mice[40] (Jackson Laboratory) to produce *Ins1$^{CreERT2}$;Swell1$^{fl/fl}$* mice. Offsprings were genotyped by PCR using *Ins1$^{CreERT2}$* and *Swell1* floxed allele-specific primers:

*Ins1$^{CreERT2}$*-F: 5′-TGGACTATAAAGCTGGTGGGCAT-3′
*Ins1$^{CreERT2}$*-R: 5′-TGCGAACCTCATCACTCGT-3′
*Swell1$^{fl}$*-F: 5′-CTAATCAGGGAGAGACAGCAGAG-3′
*Swell1$^{fl}$*-R: 5′-GATAGTTCTGGCCAGTGAGTGG-3′

To achieve tamoxifen-induced β-cell-targeted *Swell1* recombination, *Ins1$^{CreERT2}$;Swell1$^{fl/fl}$* mice (6–8-week-old males), were treated by oral gavage with tamoxifen 40–80 mg/kg (Sigma-Aldrich, T6648) dissolved in sunflower seed oil (Sigma-Aldrich, S5007) five times over a 2-week period. Littermate, gender-matched *Swell1$^{fl/fl}$* mice treated with tamoxifen as above served as controls (WT). To test for recombination efficiency *Ins1$^{CreERT2}$;Swell1$^{fl/fl}$* mice were then crossed with *Rosa26-tdTomato* mice to generate *Ins1$^{CreERT2}$;Rosa26-tdTomato;Swell1$^{fl/fl}$* mice. Mice were studied at least 4 weeks after the last tamoxifen dose to allow time for any non-specific effects of tamoxifen to wash-out. All mice used were maintained on a C57BL/6 background.

**Confocal microscopy**. To assess recombination efficiency of *Ins1$^{CreERT2}$* mice, we isolated islets from tamoxifen-induced and untreated *Ins1$^{CreERT2}$;Rosa26-tdTomato;Swell1$^{fl/fl}$* mice and imaged freshly isolated islets with a Zeiss LSM510 confocal laser scanning microscope (Zeiss, NY, USA) using a 20x/0.8 NA objective lens. Islets were excited by 561 nm light and emission signals were recorded at 581 nm in a tile mode (5 × 5) in order to observe the tdTomato fluorescence in larger fields of view. Then all 25 individual frames were tiled together to form the complete image. All images were processed and analyzed with ZEN 2009 (Zeiss) software.

**Glucose tolerance test**. Mice were deprived of food for 6 h and then injected with glucose (intraperitoneal, 2 g/kg body weight). Blood was then collected via the tail at 0, 5, 15, 30, 60, 90, and 120 min after the glucose injection. For HFD mice, 1 g/kg glucose was injected (i.p.). Glucose levels in the collected blood were determined by blood glucose meter (Bayer HealthCare LLC).

**In vivo insulin secretion assay**. For in vivo measurements of GSIS, tamoxifen-induced *Swell1$^{fl/fl}$* and *Ins1$^{CreERT2}$;Swell1$^{fl/fl}$* mice (littermate, gender-matched) were fasted 6 h prior to the experiments. Blood samples were collected from the tail vein with microvette (Sarstedt,16.444.100) before the injection of glucose as 0 min time point. Mice were then injected with 2 g/kg (BW) glucose (i.p.). Blood samples were collected from the tail vein at 7, 15, and 30 min time points and centrifuged for plasma collection (2000 × *g*, 20 min, 4 °C). Plasma insulin concentrations were determined by using an ELISA kit (Crystal Chem Inc., 90080).

**Statistics**. Data are represented as mean ± s.e.m. Two-tail paired or unpaired Student's *t*-tests were used for comparison between groups. For three or more groups, data were analyzed by one-way analysis of variance and Tukey's post hoc test. For glucose tolerance test, data were analyzed by two-way analysis of variance and Tukey's post hoc test. A probability of *p* < 0.05 was considered to be statistically significant.

**Data availability**. All relevant data, within reason, will be available from the authors on request.

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

## Acknowledgements

We thank Dr John Engelhardt for sharing human islets obtained from the Integrated Islet Distribution Program (IIDP) and Dr Yumi Imai for sharing human islets obtained from Prodo Laboratories. We thank Shanming Hu for assisting with mouse islet isolations. We thank Dr Ahmad Alghanem for assistance with figure illustrations and Dr Robert Tsushima for thoughtful reading of the manuscript and comments. This work was supported by grants from the NIH NIDDK 1R01DK106009 (R.S.), AHA Grant-in-aid 17GRNT33700001 (R.S.), R01DK097820 (A.W.N.), R24DK96518 (A.W.N.), and the Roy J. Carver Trust (R.S.).

## Author contributions

Conceptualization, R.S.; methodology, C.K., S.B.S., A.W.N., A.M., Y.Z., L.X., S.K.G., S.P., Y.G., A.K., and R.S.; formal analysis, C.K., L.X., and R.S.; investigation, C.K., S.B.S., A.M., S.P., Y.G., and A.K.; resources, R.S. and A.W.N.; writing – original draft, R.S., C.K., and L.X.; writing – review and editing, R.S., C.K., L.X., S.B.S, and A.W.N.; visualization, C.K., L.X., A.K., and R.S.; supervision, R.S.; funding acquisition, R.S.

## Additional information

**Competing interests:** The authors declare no competing financial interests.

