## [Peer Review File · Nature Communications]

Reviewer #1 (expert in beta-cell function, insulin secretion) (Remarks to the Author):

Chen Kang et al have examined the role of the recently described cell surface channel component SWELL (LRRC8a) in pancreatic beta cells using in vitro as well as in vivo models.

The presented studies have been conducted carefully with state of the art methods and mostly with appropriate controls. The data is adequately displayed.

Murine insulinoma Min6 cells, murine primary beta cells as well as human beta cells show swell (experimental exposure to hypotonic solution)-activated VRAC. This VRAC is significantly reduced upon knockdown of SWELL1.

Murine insulinoma Min6 cells in perforated patch clamp studies show VRAC after exposure to high glucose levels (30 mM)

Exposure of murine and human beta cells show increase in size of beta cells by approximately 6-8 % over the course of 12 minutes during exposure to elevated glucose levels (16.7 mM) followed by a volume reduction (regulatory volume decrease (RVD) this RVD is absent under circumstances when SWELL is knocked down using an shRNA delivery method.

Similar to Min6 cells perforated patch clamp studies reveal VRAC after glucose exposure (16 mM), which is present in WT conditions and significantly muted after SWELL1 knockdown.

Measurements of calcium transients using FURA2 in Min6 cells show accordingly that glucose as well as exposure to hypotonic solution are diminished after SWELL KO whereas exposure to KCL stimulates calcium transients equally in the setting of SWELL absence or presence.

In mouse and human primary islet beta cells, glucose stimulated calcium transients are also significantly diminished when SWELL1 is knocked down, while KCL dependent Ca transients are not significantly diminished.

Consistent with the above observations, glucose stimulated insulin secretion from Min6 cells is reduced in the setting of SWELL KO while cellular insulin content is not affected by SWELL KO. Similarly, mouse islets show reduced GSIS after β -cells specific SWELL KO, while arginine stimulation shows non-significant reduction in insulin secretion after SWELL KO.

Finally, in mice with tamoxifen (CRE-Ert/LoxP method) inducible β -cell selective SWELL knockdown, glucose stimulated insulin levels are diminished at 7 and 30 minutes as compared to control conditions.

Several concerns are noted. Addressing these would improve the interpretation of the data and address doubts and possible alternative mechanisms that may underlie the findings.

The main difficulty of this work is the interpretation that SWELL is a "glucose sensor". This is at best an overstatement of the presented data. Careful dose response studies with glucose concentrations in the physiologic range (see comments below) would be required to establish whether SWELL is a glucose sensor and how SWELL function may relate to the canonical glucose sensing mechanism that is assigned to glucokinase and its regulatory mechanisms.

Experimental glucose levels that are used for Min6 cells are adequate but do not represent physiologic concentrations as the authors claim (e.g. lines 76/77)

Similarly glucose levels of 16.7 mM in mouse and human beta cells are mainly in the profoundly diabetic range (300 mg/dl) and hardly ever reached in physiologic circumstances in mice or humans. The interpretation of these findings is therefore quite difficult as they may relate to physiologic functioning of pancreatic beta cells. Thus the swelling may be of little relevance in normal beta cell function but may be important in the context of diabetes mellitus. Do the authors

have data generated lower glucose levels, which may be of greater physiologic relevance (e.g. 7mM, 10 mM)?

Further, the authors should comment whether the observed swelling represents physiologic circumstances or observations made under extreme un-physiologic experimental conditions.

In this context related to volume changes in β -cells upon glucose exposure, the authors may be well served to examine literature related to the role of aquaporins in beta cell function at physiologic glucose levels as these also influence swelling, albeit without clear documented involvement in generating VRAC

The studies on calcium transients suggest that the SWELL-dependent function of stimulus-secretion coupling occurs upstream of the KATP/SUR1 complex. Consequently proper functioning of the KATP/SUR1 channel would be necessary for the SWELL-dependent cascade towards insulin granule exocytosis to properly proceed. This would require glucose metabolism and cellular increase in [ATP] (via glycolysis and oxidative phosphorylation). The authors should examine whether ATP production in SWELL WT and SWELL KO/KD beta cells remains intact or is impacted. NAD/NADH ratio may serve as a surrogate (albeit suboptimal) for this measurement. Similarly, sulfonylurea treatment of cells and mice followed by observation of calcium transients, insulin secretion and in vivo glucose homeostasis may reveal additional information on the relationship between SWELL1 and Katp/SUR1.

The source of Ins-CRE-ERT mice needs to be carefully established. The available mice generated from Dr. Philipson's laboratory contain a Growth hormone minigene which can influence beta cell behavior. This has been reported in a recent publication in Cell Metabolism. To address his concern, the authors should include additional controls of INS-CRE-ERT/SWELL wt/wt (ie unfloxed) mice for their controls in relevant studies (Figure 4F etc)

Figure 4J: while glucose stimulated insulin secretion is presented for 0, 7, and 30 minutes, the in vivo glucose levels are not presented. This data should be included for adequate interpretation of the data with respect to the physiologic relevance of the findings.

Reviewer #2 (expert in Volume-Regulated Anion Channels)(Remarks to the Author):

Insulin secretion by pancreatic β -cells is stimulated by an increase in plasma glucose concentration, a process known as stimulus-secretion coupling. A major aspect of such stimulation is the induction of β -cell electrical activity. The current "consensus model" proposes that glucose metabolism increases the intracellular ATP/ADP ratio, which leads to the closure of ATP-sensitive K⁺ channels (KATP), thereby inducing membrane depolarization. This in turn opens the voltage-gated Ca²⁺ channels, leading to Ca²⁺ entry into the cell, thus triggering insulin release by exocytosis. There is no doubt that KATP channels play a key role in insulin secretion; however, strong evidence indicates the existence of additional ionic pathways sensitive to increased glucose level. Previous pharmacological and electrophysiological studies suggest that Volume-Regulated Anion Channel (VRAC) may constitute additional KATP-independent glucose-sensing mechanism. Recent breakthrough of VRAC molecular cloning (SWELL1/LRRC8a) by Qiu et al and Voss et al finally made it possible to test this hypothesis which is addressed in the current manuscript. Through a series of elegant in vitro and in vivo experiments on human and mouse islets/cells, the authors concluded that SWELL1-mediated "swell-secretion" coupling is essential for both glucose-stimulated calcium increase and insulin secretion. This is a novel and significant finding which would establish a critical role for the swelling-activated chloride channel in β -cell insulin secretion. Considering that many ion channels (cation channels, such as TRP channels; chloride channels such as CFTR and TMEM16s) are implicated in KATP-independent β -cell depolarization, three main

points are discussed below which would strengthen the main conclusion from the current study.

1) The nature of glucose-induced currents in Figure 2 is suggested to be VRAC based on the delayed timing and DCPIB blocking. More evidence is necessary for this claim. DCPIB is a potent VRAC blocker and typically starts to inhibit VRAC currents in seconds. In Figure 2c and 2d, DCPIB first potentiates and then inhibits VRAC in minute-scale. In addition, DCPIB is known to inhibit other ion channels. It would be helpful to record the available SWELL1 knockout or knockdown mouse and human β -cell cells. The glucose-stimulated currents are quite small based on the examples (5-10 pA/pS at +100 mV), compared to the typical swelling-activated currents (\sim 100 pA/pS). A quantitative analysis on the current densities at \pm 100 mV and their reversal potentials would be helpful to determine its property (for example, is it mediated by Cl^- ?). Conceptually, one needs inward currents (Cl^- leaving cells) to achieve glucose-stimulated β -cell depolarization. However, the glucose-induced inward currents are minimal above the background. The author may want to discuss how this channel characteristic in β -cell (i.e. strong outward rectification) may affect its physiological function in insulin secretion.

2) In Figures 3 and 4, the authors present exciting data that knocking out SWELL1 alone completely abolishes the glucose-stimulated Ca^{2+} entry and insulin secretion. It's an extraordinary finding because it's generally agreed in β -cell literature that the membrane depolarization of β -cell is a cooperative act of the closure of KATP and opening of others such as Cl^- channel and other cation channels. It would be helpful to support the author's conclusion by investigating the role of other players (and their potential collaboration with SWELL1) such as the key KATP in their experimental system. Temporal resolution of membrane potential change and insulin secretion would be essential to establish the precise role of SWELL1. For example, in Figure 3c, 30 mM glucose stimulates Ca^{2+} entry with little delay, but it only activated very modest VRAC in 10 min (Figure 2).

3) In Figure 4, the authors show that β -cell-specific SWELL1 knockout mice completely lacks glucose-stimulated insulin secretion at both 7 min and 30 min. This provides important support for its role in vivo. More detailed analysis of the mouse model is necessary. What is the physiological consequence of this severe insulin secretion deficits, such as their glucose tolerance? Is the cre line truly β -cell-specific given the problems with various delete lines? Is SWELL1 required for β -cell/islet health and morphology, which may contributes to the strong phenotype?

Reviewer #3 (expert in Ca and K channels in islets)(Remarks to the Author):

Summary of key results: The authors studied the role of SWELL1 (LRRC8a) in beta-cell function. SWELL1 is considered as a mediator of swell-activated Cl^- current. They show that SWELL1 mediates a volume-regulated anion current (VRAC) in beta-cells. VRAC is activated by glucose-induced swelling, and this current is inhibited by a VRAC inhibitor, DCPIB. Because this current is depolarizing in beta-cells, they suggests that it could be involved in the depolarizing effect of glucose. They indeed show that glucose and hypotonic medium loose they capacity to increase $[\text{Ca}^{2+}]_i$ in SWELL1 KO beta-cells. Consequently, SWELL1 KO beta-cells loose also their ability to secrete insulin in response to glucose.

Originality: the study is very original.

Data and methodology: most of the experiments are logic. The experimental approaches are original, technically pertinent and of high quality.

Comments:

- There is little introduction about the different types of VRACs and the specific role of SWELL1 among the VRACs. Also, is SWELL1 responsible of a RVD only?
- Is LRRC8a required for VRAC? Can the other LRRC8b-e mediate a VRAC in the absence of LRRC8a?
- What are the expressions of LRRC8a-e isoforms in beta-cells? What are their relative

abundances?

- At the end of p3, the authors state that the current they measure is a Cl current. But they did not measure the selectivity. Is SWELL1 also permeable to aspartate, glutamate or other anions? What is the selectivity?
- The experiments of volume measurements (fig 2A-B) are not very convincing. Is the method used good enough? The authors used GcCaMP6 which is Ca sensitive. But if the fluorescence increases in response to glucose, it might produce an artefactual change in apparent cell volume (because pixels of the image at the edge of the cell become illuminated by the higher intensity of fluorescence). They should have used a probe with no change in fluorescence. In addition, the period before glucose stimulation is too short.
- It would have been nice to make also volume measurements in SWELL1 KO cells.
- It is very striking that KO of SWELL1 fully prevents the effect of glucose on $[Ca^{2+}]_i$ but does not affect the effect of high K (Fig 3). Does the effect of glucose in WT cells depend on glucose metabolism? What are the mechanisms by which glucose would affect SWELL1? Is it possible that KO of SWELL1 has fully impaired the metabolism of glucose. The authors should have tested whether glucose-induced change in cell metabolism (NAD(P)H fluorescence, or ATP concentration, ..) is affected by the KO of SWELL1. In addition, does KO of SWELL1 also affect the effect of other nutrients? What is the effect of sucrose?
- An outflow of Cl from the cell is expected to depolarize the cell. It is also expected to decrease cell volume. Why then does glucose increase cell volume?
- Since the effect of glucose depends on the closure of KATP channels, the authors should have tested the effect of KATP channel blockers. What are they doing on VRAC and $[Ca^{2+}]_i$ in WT and SWELL1 KO cells?
- KO of SWELL1 impairs glucose-induced insulin secretion. Does it affect high K induced insulin release?
- What does Fig4J represent? Insulinemia during IPGTT? If this is the case, it should not be presented as "insulin secretion". Moreover, how is it possible that the mice are not diabetic? Are they more prone to diabetes when submitted to a diabetogenic condition (such as high fat-diet)?
- How do the authors reconcile their data with those of Eberhardson et al Cellular Signalling 12 (2000) 781-86, in which it was found that a change in the glucose concentration does not affect $[Cl^-]_i$ in beta-cells?

Minors:

- Several figs (such as Fig 1F and J) show evolution of currents with time. The voltage at which the cell was hold is not clearly indicated.

Conclusion: the study is very original and provocative. It is potentially very interesting for our comprehension of the mechanisms by which glucose controls beta-cells, but some additional experiments are required to strengthen the data and the interpretations.

Responses to Reviewers' Comments

General responses:

We thank all reviewers for their comments. We feel that addressing these comments, in most cases with new experiments, has considerably improved our manuscript and strengthened our findings. While the Reviewers each had specific concerns (which we address in point by point detail below) there were two main overarching concerns that most Reviewers shared in common, and that were highlighted in the Editor's Letter. The *first* related to providing stronger experimental evidence that SWELL1 behaves as a glucose-sensor (Reviewers #1&2) and the *second* related to investigating further the contribution of KATP channels (Reviewers 1-3):

“In particular we ask you to provide compelling evidences that SWELL1 is a glucose sensor (as suggested by reviewer #1), and to investigate the possible contribution of KATP channels as suggested all three the reviewers.”

To address the first point about SWELL1-mediated VRAC activating in response to glucose-stimulation and thus acting as a glucose sensor, we have performed additional patch-clamp experiments (in perforated-patch configuration) as suggested by Reviewer 2, now in *both* WT *and* SWELL1 KO primary mouse β -cells (**Figure 2**). In our previous version, we had used DCPIB application (a selective VRAC/ $I_{CL,SWELL}$ inhibitor) to demonstrate glucose-stimulated activation of SWELL1-mediated $I_{CL,SWELL}$ in β -cells. However, although suggestive, this did not prove that this glucose-stimulated current was SWELL1-mediated. In the revised manuscript, we now show that this glucose-stimulated VRAC-like (or $I_{CL,SWELL}$ -like) outwardly-rectifying current is both DCPIB-sensitive and entirely absent in SWELL1 KO β -cells, which proves that SWELL1 is required for this glucose-stimulated current. Moreover, we would like to emphasize that glucose-stimulated β -cell membrane depolarization (**Figure 3**), glucose-stimulated β -cell Ca^{2+} signaling (**Figure 4&5**), glucose-stimulated insulin secretion (*in vitro* and *in vivo*, **Figures 7&8**) and glucose tolerance (*in vivo*, **Figure 8**) are *all significantly and specifically* impaired with β -cell targeted SWELL1 loss-of function (SWELL1 KO β -cells/islets are still responsive to KCl and L-arginine stimulation). In our current working model, SWELL1-mediated VRAC is glucose-activated *via glucose-mediated β -cell swelling*, as proposed by several authors previously¹⁻⁶. Therefore, we stand by our description of SWELL1 *acting as a glucose-sensor*, but in the current manuscript we have modified our description to describe SWELL1 as a “*virtual glucose sensor*” since, based on our current data, we are not suggesting that the SWELL1 channel complex is directly sensing glucose, like glucokinase as Reviewer #1 correctly points out. Instead, we propose that SWELL1-mediated $I_{CL,SWELL}$ activates in response to the glucose-stimulated β -cell swelling that occurs as a result of glucose metabolism (which is glucokinase-dependent). We hope that this additional data and modified language will clarify this point.

To address the second point regarding the contribution of KATP channels we have included additional experiments (**Figure 6**) and a section in the Results that specifically examines the relationship between depolarizing SWELL1-mediated $I_{CL,SWELL}$ and hyperpolarizing K(ATP) channels, entitled: “**The balance of SWELL1-mediated $I_{CL,SWELL}$ and K(ATP) regulates β -cell**

Ca²⁺ signaling". We used a pharmacological approach to selectively modulate Cl⁻ current through I_{Cl,SWELL} versus K(ATP) channel activity to parse the relative contributions of each to membrane excitability. Specifically, we show that inhibition of the NKCC1 co-transporter with bumetanide, and consequent reduction in β-cell intracellular Cl⁻ current is sufficient to inhibit glucose-stimulated Ca²⁺ transients, despite intact K(ATP) closure - demonstrating the importance of a glucose-stimulated I_{Cl,SWELL} for β-cell excitation. Furthermore, we show that sustained activation of K(ATP) using diazoxide is sufficient to suppress glucose-stimulated Ca²⁺ transients, despite intact SWELL1-mediated I_{Cl,SWELL} activation – demonstrating the importance of K(ATP) closure for β-cell excitation. By applying increasing concentrations of glibenclamide (0.25 nM to 10 nM) we are able to demonstrate that SWELL1-mediated I_{Cl,SWELL} (depolarizing) and K(ATP) (hyperpolarizing) counterbalance each other to regulate β-cell excitability in response to glucose (**Figure 6**).

In addition to the points described above, we have made additional significant additions to the manuscript (expanding from a 4 figure paper to 9 figures), including:

1. Directly measuring β-cell membrane potential in current-clamp mode in both whole-cell and perforated patch configurations (at 37°C) in response to hypotonic swelling and glucose-stimulation, respectively (**Figure 3**) (Raised by Reviewer 2). These data show that SWELL1 is indeed required for normal β-cell membrane depolarization in response to both hypotonic swelling and glucose-stimulation.
2. Demonstrating that a non-metabolizable form of glucose, 3-O-methylglucose, is ineffective at stimulating β-cell Ca²⁺ transients (**Figure 6a**), as shown previously⁵ (Raised by Reviewer 3), emphasizing the importance of glucose metabolism for β-cell stimulation.
3. We have extensively expanded our *in vivo* characterization to now include glucose-tolerance of *Ins1^{CreERT2};SWELL1^{fl/fl}* and control mice *pre* and *post* tamoxifen-mediated SWELL1 deletion to clearly demonstrate β-cell SWELL1 dependent glucose-intolerance (**Figure 8e-f**). In a separate cohort of mice, we demonstrate that this glucose-intolerance arising from impaired glucose-stimulated insulin secretion in mice with SWELL1 deficient β-cells is further exacerbated in the setting of mild obesity in HFD fed mice for 5-6 weeks (**Figure 8i-l**). These new data highlight both the physiological and pathophysiological significance of β-cell SWELL1 (Raised by all Reviewers 1, 2 and 3), and as mentioned in the Discussion, suggest that impaired SWELL1 activity might drive Type 2 DM.

Responses to Reviewer #1:

“The main difficulty of this work is the interpretation that SWELL is a “glucose sensor”. This is at best an overstatement of the presented data. Careful dose response studies with glucose concentrations in the physiologic range (see comments below) would be required to establish whether SWELL is a glucose sensor and how SWELL function may relate to the canonical glucose sensing mechanism that is assigned to glucokinase and its regulatory mechanisms.”

We thank the Reviewer for this comment. Our intention was to present SWELL1-mediated $VRAC/I_{Cl,SWELL}$ as a sensor of β -cell swelling - a reasonable assertion based on our data and others^{2,3,6-9}. It is this β -cell swelling that occurs in response to glucose stimulation and requires glucose metabolism as shown by others previously⁵. In this way, SWELL1 can be conceived of as a *virtual glucose sensor*, since its activation is coupled to glucose-dependent processes (i.e. glucose metabolism) and consequent β -cell swelling⁵. Indeed, glucose metabolism is dependent on glucokinase, which directly senses glucose, as the Reviewer points out. It was certainly not our intention to suggest that SWELL1 is *directly* sensing glucose, but instead is indirectly connected to a glucose-sensitive processes – namely β -cell swelling^{1,2,5,6}. Therefore, we have changed our description of SWELL1 to a “virtual glucose sensor” as our data is certainly consistent with this notion (**See General Responses**).

“Similarly glucose levels of 16.7 mM in mouse and human beta cells are mainly in the profoundly diabetic range (300 mg/dl) and hardly ever reached in physiologic circumstances in mice or humans. The interpretation of these findings is therefore quite difficult as they may relate to physiologic functioning of pancreatic beta cells. Thus the swelling may be of little relevance in normal beta cell function but may be important in the context of diabetes mellitus. Do the authors have data generated lower glucose levels, which may be of greater physiologic relevance (e.g. 7mM, 10 mM)? Further, the authors should comment whether the observed swelling represents physiologic circumstances or observations made under extreme un-physiologic experimental conditions.”

The use of 16.7 mM stimulatory glucose is quite standard in the field and not typically perceived of as unphysiological, particularly in β -cell lines and in primary β -cells from rodents^{1,5,10}, although there is evidence that human β -cells may be more glucose-sensitive¹¹. Indeed, serum glucose values in the 300 mg/dl range are commonly encountered during standard glucose-tolerance testing (GTT) in lean, glucose-tolerant mice (**Figure 8e, f&i**)¹². We agree that 16.7 mM glucose (300 mg/dl) is certainly high for fasting glucose, but this is not what we are testing in these experiments. We are showing that glucose-stimulation of the pancreatic β -cell - as occurs with feeding (not fasting) - requires β -cell SWELL1. In fact, we find that basal (analogous to fasting) insulin secretion *in vitro* (**Figure 7**), fasting serum insulin *in vivo* (**Figure 8g**) and fasting glucose *in vivo* (**Figure 8e, f, i**) are identical in WT compared to β -cell SWELL1 KO islets and mice – supporting the notion that SWELL1 regulates insulin secretion in the context of a nutrient load (glucose) under non-pathological conditions. Interestingly, with very mild obesity and associated mild insulin resistance, β -cell SWELL1 deficiency does induce fasting hyperglycemia (**Figure 8j**), suggesting that SWELL1 may be required to maintain normoglycemia in the setting of obesity. Also, since with the standard 16.7 mM glucose stimulation, SWELL1 KO β -cells are essentially non-responsive, it made little sense to test lower glucose concentrations in our *in vitro* experiments.

“The studies on calcium transients suggest that the SWELL-dependent function of stimulus-secretion coupling occurs upstream of the KATP/SUR1 complex. Consequently proper functioning of the KATP/SUR1 channel would be necessary for the SWELL-dependent cascade

towards insulin granule exocytosis to properly proceed... , sulfonyleurea treatment of cells and mice followed by observation of calcium transients, insulin secretion and in vivo glucose homeostasis may reveal additional information on the relationship between SWELL1 and KATP/SUR1.”

We thank the Reviewer for raising the question about the relationship between SWELL1 and KATP/SUR1. This was a common question among the Reviewers and thus is addressed in **General Responses**, above. Briefly, we addressed this comment by examining glucose-stimulated Ca^{2+} signaling in the presence of K(ATP) modulators (glibenclamide/diazoxide) to demonstrate that it is the balanced activity of depolarizing SWELL1-mediated Cl^- current and hyperpolarizing K(ATP) current that ultimately regulates β -cell excitability (**Figure 9**).

“The source of Ins-CRE-ERT mice needs to be carefully established. The available mice generated from Dr. Philipson’s laboratory contain a Growth hormone minigene which can influence beta cell behavior. This has been reported in a recent publication in Cell Metabolism. To address his concern, the authors should include additional controls of INS-CRE-ERT/SWELL wt/wt (ie unfloxed) mice for their controls in relevant studies (Figure 4F etc)”

We appreciate the Reviewer’s concern regarding the mice generated in Dr. Philipson’s laboratory. Fortunately, we did not use these mice, but instead used $\text{Ins1}^{\text{CreERT2}}$ mice generated by Dr. Ferrer as described in Thorens et al.¹³ and available from Jackson Labs (<https://www.jax.org/strain/026802>). These mice do not have the above mentioned issues and have also been previously characterized to exhibit no baseline effects on growth and glucose metabolism¹³. Accordingly, we have not generated the additional controls suggested by the Reviewer. We do thank the Reviewer for pointing out that we need to more clearly indicate the source of the mice and we have amended our manuscript accordingly. Moreover, in our revised manuscript we do include an experiment that entirely rules out the possibility of a “Cre line effect” by examining glucose tolerance in mice pre and post tamoxifen-induction (**Figure 8e&f**). In this experiment, the $\text{Ins1}^{\text{CreERT2}}; \text{SWELL1}^{\text{fl/fl}}$ mice have identical glucose tolerance to $\text{SWELL1}^{\text{fl/fl}}$ littermates prior to tamoxifen induction, but significant dysglycemia post tamoxifen induction. Also, this experiment rules out the possibility of any developmental effects on β -cell differentiation or maturation as a cause for the phenotypes observed *in vivo*.

“Figure 4J: while glucose stimulated insulin secretion is presented for 0, 7, and 30 minutes, the in vivo glucose levels are not presented. This data should be included for adequate interpretation of the data with respect to the physiologic relevance of the findings.”

We entirely agree with the Reviewer on this point and in the revised manuscript we now present a detailed evaluation of glucose tolerance in both lean and HFD fed mice (**Figure 8**). Please see **General Responses**.

Responses to Reviewer #2:

“The nature of glucose-induced currents in Figure 2 is suggested to be VRAC based on the delayed timing and DCPIB blocking. More evidence is necessary for this claim. DCPIB is a potent VRAC blocker and typically starts to inhibit VRAC currents in seconds. In Figure 2c and 2d, DCPIB first potentiates and then inhibits VRAC in minute-scale. In addition, DCPIB is known to inhibit other ion channels. It would be helpful to record the available SWELL1 knockout or knockdown mouse and human β -cell cells.”

We agree with the Reviewer on this point, and as described above in **General Responses**, in the revised version of the manuscript we now include data demonstrating that the glucose-stimulated DCPIB-sensitive current is entirely absent in SWELL1 KO murine β -cells. This data now proves this current to be SWELL1 dependent. We suspect that the apparent potentiation of VRAC and then inhibition upon DCPIB application in Figure 2c and 2d of the previous version of the manuscript (human β -cells, **Supp. Fig 2c** in current version) is related to the fact that glucose was stimulating VRAC during DCPIB application. On occasion, under perforated patch at 37°C, DCPIB can take up to a couple of minutes to start to block VRAC under glucose stimulation in β -cells. This longer latency for DCPIB inhibition may be related to the conditions used for these recordings (37°C and perforated patch), as most DCPIB inhibition of VRAC/ $I_{CL,SWELL}$ are performed at RT in whole-cell mode, but even in these cases significant VRAC/ $I_{CL,SWELL}$ inhibition can take several minutes¹². Indeed, current-clamp recordings by Best et al. (2004) under similar conditions to ours also show that DCPIB mediated $I_{CL,SWELL}$ inhibition can take several minutes to reduce β -cell excitability during glucose-stimulation (**See Fig. 5**)⁴.

“The glucose-stimulated currents are quite small based on the examples (5-10 pA/pS at +100 mV), compared to the typical swelling-activated currents (~100 pA/pS). A quantitative analysis on the current densities at +/-100 mV and their reversal potentials would be helpful to determine its property (for example, is it mediated by Cl^- ?). “

In the revised manuscript, we now provide quantitative mean data for the glucose-stimulated SWELL1-dependent current at both +100 and -100 mV in primary murine β -cells (**Figure 2f**). Through these additional experiments, we learned that the glucose-stimulated SWELL1-mediated $I_{Cl,SWELL}$ currents are indeed smaller than hypotonic swell-activated currents at +100 mV but actually trend toward larger in magnitude at -100 mV, which is the physiologically relevant range of MP responsible for β -cell depolarization: glucose: $+40.3 \pm 8.5$ pA/pF; swell: $+81.9 \pm 19.9$ pA/pF at +100 mV; glucose: -14.3 ± 3.5 pA/pF; swell: -11.0 ± 1.2 pA/pF at -100 mV. It should also be noted that there are important differences in experimental conditions between the hypotonic swell-activated recordings compared to the glucose-stimulated recordings. Hypotonic swell recordings are performed at RT and in whole-cell configuration, while glucose-stimulated recordings must be performed under more technically challenging physiological conditions (35-37°C, and perforated patch configuration), and this may certainly alter $I_{Cl,SWELL}$ characteristics. Indeed, most of the published literature on VRAC/ $I_{Cl,SWELL}$ perform recordings in whole-cell mode and at RT. Our data suggest that this approach, while easier, will probably miss important physiologically relevant features and functions of this ionic current.

Another important detail is that, under perforated-patch configuration, we are unable to control the intracellular ionic concentrations, and thus additional background currents (i.e. monovalent cations) can also be present in the recordings. Nonetheless, the *SWELL1* KO β -cells show that *SWELL1*-mediated current is dominant and significant, even under perforated patch conditions.

With respect to identifying the *SWELL1*-mediated current as a Cl^- current, in the Results section we do note that the reversal potential for the hypotonic swell-activated current is -12 mV which is close to the expected reversal potential for Cl^- under our conditions, and thus consistent with a Cl^- conductance. While, the reversal potential of glucose-stimulated *SWELL1* current under perforated patch conditions (at 35-37°C) does appear right-shifted compared to whole-cell conditions, this may reflect the additional background currents described above. Moreover, this glucose-stimulated current is DCPIB sensitive (a selective $I_{\text{Cl},\text{SWELL}}$ inhibitor), is abolished upon *SWELL1* deletion (an established ion channel component necessary for $I_{\text{Cl},\text{SWELL}}$ in other cell types^{7, 8, 12}) and has been observed in β -cells by others previously to be mediated by Cl^- ^{1, 3, 4, 9}. Therefore, collectively, we feel that there is ample evidence that *SWELL1* mediates a glucose-sensitive Cl^- conductance in β -cells.

“Conceptually, one needs inward currents (Cl^- leaving cells) to achieve glucose-stimulated β -cell depolarization. However, the glucose-induced inward currents are minimal above the background.”

Our revised data indicate that the glucose-stimulated *SWELL1*-mediated inward Cl^- current in murine β -cells is -14.3 ± 3.5 pA/pF at -100 mV. Bearing in mind that this inward depolarizing current (i.e. “accelerator”) activates concurrently with closure of hyperpolarizing K(ATP) (i.e. removal of the “brake”), it is reasonable that it may contribute to glucose-stimulated β -cell depolarization, as it may not take much depolarizing current to “tip the balance” toward membrane depolarization as K(ATP) channels close. Moreover, in **Figure 3** we now present current-clamp data to show that both glucose and hypotonic swelling induced β -cell membrane depolarization has a significant *SWELL1*-dependent contribution.

“The author may want to discuss how this channel characteristic in β -cell (i.e. strong outward rectification) may affect its physiological function in insulin secretion.”

The Reviewer raises an interesting point. The property of outward rectification will result in *SWELL1*-mediated Cl^- contributing more strongly to β -cell membrane depolarization at hyperpolarized potentials, thereby initiating β -cell excitation, but then “shuts-off” at more depolarized potentials to allow for requisite β -cell repolarization and termination of insulin secretion. Moreover, as the membrane potential exceeds the equilibrium potential of Cl^- ($E_{\text{Cl}^-} = \sim -15$ mV), *SWELL1*-mediated $I_{\text{Cl},\text{SWELL}}$ will provide a hyperpolarizing current to stabilize the β -cell around E_{Cl^-} , thereby maximizing Ca^{2+} influx via voltage-gated Ca^{2+} channels. We have added this point to our Discussion section. Indeed, this concept has been alluded to in the literature by Eberhardson et. al. Cellular Signalling 12 (2000) 781-86¹⁴ : “The present study provides ample support for the proposal [5] that Cl^- accumulates in pancreatic β -cell against its electrochemical

gradient and that glucose stimulates the efflux of the ion. Being electrogenic, the exit of Cl⁻ adds to the depolarizing action of glucose and consequently facilitates the influx of Ca²⁺ + responsible for the rising phase of the [Ca²⁺]_i oscillations. However, when the β-cell become sufficiently depolarized, the electrochemical gradient may drive Cl⁻ in the opposite direction, thereby stabilizing the membrane potential.” We thank the Reviewer for suggesting this and we now discuss this in the Discussion of the revised manuscript.

“In Figures 3 and 4, the authors present exciting data that knocking out SWELL1 alone completely abolishes the glucose-stimulated Ca²⁺ entry and insulin secretion. It’s an extraordinary finding because it’s generally agreed in β-cell literature that the membrane depolarization of β-cell is a cooperative act of the closure of K(ATP) and opening of others such as Cl⁻ channel and other cation channels. It would be helpful to support the author’s conclusion by investigating the role of other players (and their potential collaboration with SWELL1) such as the key K(ATP) in their experimental system.”

We agree completely with this comment and have accordingly examined the contribution the K(ATP) channel and SWELL1 currents through pharmacological approaches (See **General Responses**). Briefly, we addressed this comment by examining glucose-stimulated Ca²⁺ signaling in the presence of K(ATP) modulators (glibenclamide/diazoxide) to demonstrate that it is the balanced activity of depolarizing SWELL1-mediated Cl⁻ current and hyperpolarizing K(ATP) current that ultimately regulates β-cell excitability. Thus, our current model is that SWELL1-LRRC8 and K(ATP) channels act antithetically to regulate β-cell membrane potential (**Figure 9**).

“Temporal resolution of membrane potential change and insulin secretion would be essential to establish the precise role of SWELL1.”

We agree and have now added measurements of β-cell membrane potential in current-clamp mode (**Figure 3**) to directly measure the contribution of SWELL1-mediated current to β-cell membrane depolarization.

“In Figure 4, the authors show that β-cell-specific SWELL1 knockout mice completely lacks glucose-stimulated insulin secretion at both 7 min and 30 min. This provides important support for its role in vivo. More detailed analysis of the mouse model is necessary. What is the physiological consequence of this severe insulin secretion deficits, such as their glucose tolerance?”

We agree completely and in the revised manuscript we now present a detailed evaluation of glucose tolerance in both lean and HFD fed mice (**Figure 8**) that clearly demonstrate SWELL1 dependent defects in glucose tolerance.

“Is the cre line truly β-cell-specific given the problems with various delete lines? Is SWELL1 required for β-cell/islet health and morphology, which may contributes to the strong phenotype?”

The β -cell targeted Cre line that we used, $Ins1^{CreERT2}$, has been well characterized, shown to be without off-target effects and also highly β -cell specific with efficient recombination¹³. To validate this in our hands we crossed the $Ins1^{CreERT2};SWELL1^{fl/fl}$ mice with a tdTomato reporter line. We demonstrate very efficient β -cell targeted Cre expression (SWELL1 recombination) upon tamoxifen-induction, and healthy islets with normal morphology (**Supplementary Figure 4**). Similarly, SWELL1 deletion in cultured murine and human islets is also not associated with significant morphological β -cell defects *in vitro* (**Supplementary Figure 1**). Thus, similar to our previous observations in adipocytes¹², SWELL1 is not required for β -cell survival. Instead, the strong phenotype arising from SWELL1 disruption in β -cells is entirely due to a defect in glucose-stimulated insulin secretion. This assertion is also supported by our findings that KCl induced Ca^{2+} transients are fully preserved in SWELL1 KO β -cells, that basal insulin secretion is normal in SWELL1 KO β -cells, that insulin content is preserved in SWELL1 KO β -cells, that L-arginine responses are preserved in SWELL1 KO/KD islets and that fasting serum glucose is unchanged in β -cell targeted SWELL1 KO mice raised on a regular diet.

Responses to Reviewer #3:

“- There is little introduction about the different types of VRACs and the specific role of SWELL1 among the VRACs. Also, is SWELL1 responsible of a RVD only?
- Is LRRC8a required for VRAC? Can the other LRRC8b-e mediate a VRAC in the absence of LRRC8a?
- What are the expressions of LRRC8a-e isoforms in beta-cells? What are their relative abundances?”

We agree about the lack of introduction in the earlier version of the manuscript. The previous version was a short form as submitted to another journal. We have now added an Introduction and expanded Discussion that provides more background about SWELL1/LRRC8a and associated subunits LRRC8b-e.

It is difficult to determine the expression of LRRC8a-e isoforms and relative abundances in β -cells as this required FACS sorting β -cells and then performing expression analysis, as the pancreatic islet is heterogenous and consists of several different cell types (alpha, beta, delta, epsilon cells etc). This can and will be done in future studies that combine FACS and RNA sequencing. However, our main point in this manuscript is that genetic ablation of the essential component of VRAC/ $I_{Cl,SWELL}$, SWELL1 (LRRC8a), is sufficient to disrupt VRAC/ $I_{Cl,SWELL}$ in β -cells and has significant and specific effects on glucose-stimulated β -cell excitability and systemic glucose homeostasis.

“- At the end of p3, the authors state that the current they measure is a Cl current. But they did not measure the selectivity. Is SWELL1 also permeable to aspartate, glutamate or other anions? What is the selectivity?”

Our statement that the SWELL1 mediated current is a Cl^- current is based on the -12 mV reversal potential for hypotonic swell-activated current being close to the expected reversal potential for Cl^- under our conditions (See Results), and thus consistent with a Cl^- conductance. Moreover, the glucose-stimulated current we measure is DCPIB sensitive (a selective $I_{\text{Cl},\text{SWELL}}$ inhibitor), is abolished upon SWELL1 deletion (an established ion channel component necessary for $I_{\text{Cl},\text{SWELL}}$ in other cell types^{7, 8, 12}) and has been observed by others previously to be mediated by Cl^- ^{1, 3, 4, 9}. Indeed, Best et. al. (1996) confirm this current to be Cl^- selective with a halide selectivity of $\text{Br} > \text{Cl} > \text{I}$ ³. Therefore, based on our work and the work of others, we feel that there is ample evidence that SWELL1 mediates a glucose-sensitive Cl^- conductance in β -cells.

“- The experiments of volume measurements (fig 2A-B) are not very convincing. Is the method used good enough? The authors used GCaMP6 which is Ca sensitive. But if the fluorescence increases in response to glucose, it might produce an artefactual change in apparent cell volume (because pixels of the image at the edge of the cell become illuminated by the higher intensity of fluorescence). They should have used a probe with no change in fluorescence. It would have been nice to make also volume measurements in SWELL1 KO cells.”

We believe that the Reviewer may have mis-interpreted the β -cell volume measurement experiments. In these experiments, we *did not use GCaMP6s* fluorescence to measure β -cell size but instead used bright-field microscopy (see revised Methods), since as the Reviewer correctly points out, fluorescence will create an artefactual increase in apparent cell size (albeit, the time-course of the fluorescence increase is markedly different from swelling). We did use GCaMP6s (expressed under a RIP promoter) to correctly identify the β -cell, but then performed β -cell size measurements subsequently in bright-field mode. In retrospect, we realize that we did not clearly state this in our previous version of the Methods, and for that we apologize and now appropriately make this point clear. We thank the Reviewer for drawing this omission to our attention.

Regarding volume measurements in *SWELL1 KO* cells, we did measure β -cell volume in upon SWELL1 KO/KD in both murine (**new Figure 2a**) and human (**new Supp Fig 2a**) β -cells. We find that 16.7 mM glucose induces transient β -cell swelling, followed by regulatory volume decrease (RVD) as described by others previously^{5, 6}. Notably, we find that RVD is absent in SWELL1 KO/KD β -cells.

“It is very striking that KO of SWELL1 fully prevents the effect of glucose on $[\text{Ca}^{2+}]$ but does not affect the effect of high K (Fig 3). Does the effect of glucose in WT cells depend on glucose metabolism? What are the mechanisms by which glucose would affect SWELL1? Is it possible that KO of SWELL1 has fully impaired the metabolism of glucose. The authors should have tested whether glucose-induced change in cell metabolism (NAD(P)H fluorescence, or ATP concentration, ..) is affected by the KO of SWELL1.

There are several interesting questions imbedded in this point. In the revised manuscript we show in **Figure 6a** that glucose metabolism is required since 3-O-methylglucose (non-metabolizable glucose) is ineffective at stimulating β -cells, and this is consistent with previous

work⁵. Also, we do not believe that *SWELL1* KO has fully impaired the cellular metabolism of glucose in the β -cell for a number of reasons. Full impairment of cellular glucose metabolism would probably significantly impact overall β -cell health. Indeed, β -cell targeted deletion of glucokinase (which fully impairs glucose metabolism) results in a severe perinatal lethality phenotype with severe diabetes¹⁵. This does not appear to be the case with β -cell targeted *SWELL1* KO based on preserved β -cell/islet morphology (**Supplementary Figure 4**), preserved L-arginine induced insulin secretion (**Figure 7**), preserved insulin content (**Figure 7**) and normal fasting glucose and serum insulin (**Figure 8**) in β -cell targeted *SWELL1* KO islets and mice. Importantly, normal glucose metabolism has been established to be required for β -cell swelling, as 3-O-methylglucose is incapable of inducing β -cell swelling⁵. We find that both murine *SWELL1* KO β -cells and *SWELL1* KD human β -cells undergo glucose-stimulated β -cell swelling in a manner indistinguishable from WT, albeit lacking RVD (**Figure 2a and Supplementary Figure 2a**), indicating that glucose metabolism is indeed intact in *SWELL1* KO/KD β -cells. We thank the Reviewer for raising this question as we have added a section to the Discussion to address this.

“An outflow of Cl from the cell is expected to depolarize the cell. It is also expected to decrease cell volume. Why then does glucose increase cell volume?”

We now show current-clamp data demonstrating impaired membrane depolarization in *SWELL1* KO β -cell that arises from loss of the *SWELL1*-mediated Cl⁻ current (**Figure 3**). Also, as shown in Figure 2, glucose-stimulation first causes β -cell swelling (increase in volume) due to glucose metabolism and this provides the stimulus for *SWELL1*-mediated I_{Cl,SWELL} activation. Upon activation of *SWELL1*-mediated I_{Cl,SWELL}, the β -cell depolarizes (**Figure 3**) concurrent with Cl⁻/osmolyte efflux and thus followed by RVD (**Figure 2**). Thus, the sequence is: 1. Glucose mediated β -cell swelling (increase in volume), 2. Swell-mediated *SWELL1* current activation, 3. *SWELL1*-mediated Cl⁻ induced depolarization, 4. Reduction in β -cell volume (RVD).

“Since the effect of glucose depends on the closure of KATP channels, the authors should have tested the effect of KATP channel blockers. What are they doing on VRAC and [Ca²⁺]_i in WT and *SWELL1* KO cells?”

We agree with the Reviewer on the point of studying KATP modulators, including inhibitors such as glibenclamide. This was a common question among the Reviewers and thus is addressed in **General Responses**, above. We addressed this comment by examining glucose-stimulated Ca²⁺ signaling in the presence of K(ATP) modulators (glibenclamide/diazoxide) to demonstrate that it is the balanced activity of depolarizing *SWELL1*-mediated Cl⁻ current and hyperpolarizing K(ATP) current that ultimately regulates β -cell excitability (**Figure 6**). Thus, our current model is that *SWELL1*-LRRC8 and K(ATP) channels act antithetically to regulate β -cell membrane potential (**Figure 9**).

“What does Fig4J represent? Insulinemia during IPGTT? If this is the case, it should not be presented as “insulin secretion”. Moreover, how is it possible that the mice are not diabetic? Are they more prone to diabetes when submitted to a diabetogenic condition (such as high fat-diet)?”

In the revised manuscript the *in vivo* data is presented in **Figure 8**, wherein we now show IPGTT as the Reviewer suggested, in addition to serum insulin in response to the i.p. glucose injection (**See General Responses**). Interestingly, under lean conditions, β -cell targeted *SWELL1* KO mice are not diabetic as fasting glucose is not different compared to littermate controls, however, they are significantly glucose intolerant (**Figure 8f&i**). This phenotype is entirely consistent with the notion that *SWELL1* current is glucose-stimulated and thus required for glucose-stimulated insulin secretion and not basal insulin secretion. However, as the Reviewer correctly predicted, β -cell targeted *SWELL1* KO mice are indeed prone to diabetes when submitted to a HFD for a relatively short period of time (**Figure 8j&k**).

“How do the authors reconcile their data with those of Eberhardson et al Cellular Signalling 12 (2000) 781-86, in which it was found that a change in the glucose concentration does not affect [Cl⁻] in beta-cells?”

It appears that the Reviewer may have mis-interpreted the data in the paper by Eberhardson et al Cellular Signalling 12 (2000) 781-86¹⁴; as the findings of this paper are entirely aligned and supportive of our findings and also support numerous other papers that have studied the role Cl⁻ conductance on β -cell excitability^{1, 4-6, 9}. The Reviewer specifically cites the point that “*change in the glucose concentration does not affect [Cl⁻] in beta-cells*”. Our data does not contradict these findings as we never directly measured intracellular Cl⁻ in β -cells, nor do we suggest that bulk intracellular Cl⁻ should change. Activation of a Cl⁻ channel need not necessary change the bulk intracellular Cl⁻ concentration as ion channels typically do not do this, but instead carry an electrogenic transmembrane current that either depolarizes or hyperpolarizes the plasma membrane, without changing bulk intracellular ionic concentrations (consider K⁺ channels and Na⁺ channels). Indeed, Eberhardson et. al demonstrate that Cl⁻ channel blockade with H₂DIDS does affect β -cell excitability and they conclude “*The present study provides ample support for the proposal [5] that Cl⁻ accumulates in pancreatic β -cell against its electrochemical gradient and that glucose stimulates the efflux of the ion. Being electrogenic, the exit of Cl⁻ adds to the depolarizing action of glucose and consequently facilitates the influx of Ca²⁺ responsible for the rising phase of the [Ca²⁺]_i oscillations. However, when the β -cell become sufficiently depolarized, the electrochemical gradient may drive Cl⁻ in the opposite direction, thereby stabilizing the membrane potential.*” Therefore, the paper of Eberhardson et al.¹⁴ fully supports our findings, and accordingly, there is nothing to reconcile.

References

1. Best, L. Glucose-induced electrical activity in rat pancreatic beta-cells: dependence on intracellular chloride concentration. *J Physiol* **568**, 137-144 (2005).
2. Best, L., Brown, P.D., Sener, A. & Malaisse, W.J. Electrical activity in pancreatic islet cells: The VRAC hypothesis. *Islets* **2**, 59-64 (2010).
3. Best, L., Sheader, E.A. & Brown, P.D. A volume-activated anion conductance in insulin-secreting cells. *Pflugers Arch* **431**, 363-370 (1996).
4. Best, L., Yates, A.P., Decher, N., Steinmeyer, K. & Nilius, B. Inhibition of glucose-induced electrical activity in rat pancreatic beta-cells by DCPIB, a selective inhibitor of volume-sensitive anion currents. *Eur J Pharmacol* **489**, 13-19 (2004).
5. Miley, H.E., Sheader, E.A., Brown, P.D. & Best, L. Glucose-induced swelling in rat pancreatic beta-cells. *J Physiol* **504 (Pt 1)**, 191-198 (1997).
6. Sheader, E.A., Brown, P.D. & Best, L. Swelling-induced changes in cytosolic [Ca²⁺⁺] in insulin-secreting cells: a role in regulatory volume decrease? *Mol Cell Endocrinol* **181**, 179-187 (2001).
7. Qiu, Z. *et al.* SWELL1, a Plasma Membrane Protein, Is an Essential Component of Volume-Regulated Anion Channel. *Cell* **157**, 447-458 (2014).
8. Voss, F.K. *et al.* Identification of LRRC8 heteromers as an essential component of the volume-regulated anion channel VRAC. *Science* **344**, 634-638 (2014).
9. Best, L., Miley, H.E. & Yates, A.P. Activation of an anion conductance and beta-cell depolarization during hypotonically induced insulin release. *Exp Physiol* **81**, 927-933 (1996).
10. Straub, S.G. & Sharp, G.W. Glucose-stimulated signaling pathways in biphasic insulin secretion. *Diabetes Metab Res Rev* **18**, 451-463 (2002).
11. Rorsman, P. & Braun, M. Regulation of insulin secretion in human pancreatic islets. *Annu Rev Physiol* **75**, 155-179 (2013).
12. Zhang, Y. *et al.* SWELL1 is a regulator of adipocyte size, insulin signalling and glucose homeostasis. *Nature cell biology* **19**, 504-517 (2017).
13. Thorens, B. *et al.* Ins1(Cre) knock-in mice for beta cell-specific gene recombination. *Diabetologia* **58**, 558-565 (2015).
14. Eberhardson, M., Patterson, S. & Grapengiesser, E. Microfluorometric analysis of Cl⁻ permeability and its relation to oscillatory Ca²⁺ signalling in glucose-stimulated pancreatic beta-cells. *Cell Signal* **12**, 781-786 (2000).
15. Postic, C. *et al.* Dual roles for glucokinase in glucose homeostasis as determined by liver and pancreatic beta cell-specific gene knock-outs using Cre recombinase. *J Biol Chem* **274**, 305-315 (1999).

Reviewer #1 (Remarks to the Author):

The authors have responded to all questions raised by me.

A minor point which I am sure can be readily addressed:

The Ins1 CRE-ERT mice from Thoren's laboratory - in contrast to the original publication - are noted to achieve recombination only in a minority of beta cells. Do the authors have sufficient data to support that in their hands the recombination did in fact occur in the majority of beta cells (i.e. fluorescent reporter mice activated by CRE mediated recombination)

Reviewer #2 (Remarks to the Author):

Chen Kang et. al. presented additional in vitro and in vivo data, which addressed most of my comments. However, one fundamental concern still remains regarding the role of classical KATP channel in glucose-stimulated insulin release. In the new experiments the author presented (in Fig. 6h, i, j), low dose glibenclamide closes KATP channel, thus depolarizes β -cells and stimulates Ca^{2+} influx. However, unexpectedly, this effect is dependent on SWELL1 expression. The authors speculate that under basal conditions, there is a background of constitutively active SWELL1-mediated depolarizing ICl, SWELL (at 0 mM glucose). This directly contradicts to the data and the literature that SWELL channel needs high glucose levels (i.e. 16 - 20 mM) to be active. Only under those conditions, there is presumably enough osmolarity change inside of β -cells.

In Fig 6f, g, k, l, m, the authors confirmed that KATP closure is necessary and sufficient for glucose-stimulated β -cell excitation, which is well established in the literature. However, knocking out SWELL1 alone completely abolishes the glucose-stimulated Ca^{2+} entry and insulin secretion as shown in vitro (Figs 4, 5, and 7) and in vivo (Fig. 8). Is this due to the KATP channel function compromised in SWELL1 KO cells and mice? If the KATP channel function is intact as suggested by the authors, should one expect the effects of its closure (at least partially) on glucose-stimulated membrane depolarization, Ca^{2+} entry and insulin secretion? The proposed model by Best L et. al. and the authors is that KATP channel inhibition is sensitive to glucose concentrations corresponding to hypoglycaemia (nominally 0–5 mM). VRAC channel is activated by glucose over a higher concentration range corresponding to hyperglycemia (up to 20 mM). So these two ionic mechanisms (closure of KATP and opening of VRAC) are proposed to be additive and cooperative in glucose-stimulated insulin secretion.

The cooperative act of the closure of KATP and opening of other Cl^- channels was also supported by the critical role of several other chloride channels (especially CFTR, Guo JH, Nature Communications 2014, and TMEM16A, Edlund A, BMC Medicine 2014) in modulating glucose-stimulated insulin secretion in β -cells. The authors may want to cite and discuss their findings in this context. In light of these findings, the complete lack of glucose-stimulated Ca^{2+} entry and insulin secretion in SWELL1 KO cells and mice may suggest other yet-unrealized essential functions of VRAC in β -cell biology in addition to the proposed regulation of membrane potential.

Reviewer #3 (Remarks to the Author):

The authors addressed several important issues. A few issues still need to be addressed.

- I understand that swell is not *stricto sensu* a glucose sensor. But the term "virtual" glucose sensor is confusing. I would suggest to find something else or keep "glucose sensor" as in the previous version but discussing in the paper that it this term should be qualified.

- The authors tested the effect of 3-O-methylglucose on $[\text{Ca}^{2+}]$. They show that it does not increase $[\text{Ca}^{2+}]$. This is well established. The question was whether a 3-O-methylglucose or blockade of glucose metabolism affects the SWELL1 current.

- Looking at fig 2, it is not clear at all where a RVD occurs DURING the application of glucose. 15 min after the application of glucose, there is not difference in cell volume between WT and SWELL1 KO and no sign of RVD.
- The authors did not fully address the question regarding the KATP channel modulators. Do the modulators affect the SWELL1 current?
- The authors did not answer to one request: KO of SWELL1 impairs glucose-induced insulin secretion. Does it affect high K induced insulin release?
- In fig 6b, 6d, the effect of KCl has been tested after removal of bumetanide. It should have been tested during the application of bumetanide.
- Fig. 6K, why 10 nM glib increase $[Ca^{2+}]$ with the same delay in WT than in SWELL1 KO cells?

Responses to Reviewers' Comments #2

We thank all reviewers for their additional comments and address these below:

Responses to Reviewer #1:

1. "The *Ins1* CRE-ERT mice from Thoren's laboratory - in contrast to the original publication - are noted to achieve recombination only in a minority of beta cells. Do the authors have sufficient data to support that in their hands the recombination did in fact occur in the majority of beta cells (i.e. fluorescent reporter mice activated by CRE mediated recombination)"

The Reviewer may have missed two pieces of data that quantified the recombination efficiency of *Ins1*^{CRE-ERT} mice upon tamoxifen-induction. The first is shown in Figure 8d that shows SWELL1 current amplitudes from 10 individual recordings from *Ins1*^{CreERT2};*SWELL1*^{fl/fl} mice and shows that 8/10 had essentially no $I_{Cl,SWELL}$, indicating a recombination frequency of 80%. The second, shown in Supplementary Figure 4, is precisely the experiment that the Reviewer requested. We used *Ins1*^{CreERT2};*tdTomato*;*SWELL1*^{fl/fl} induced with Tamoxifen to show high recombination efficiency in dissociated islets and no detectable recombination in un-transduced mice.

Responses to Reviewer #2:

1. "Chen Kang et al. presented additional in vitro and in vivo data, which addressed most of my comments. However, one fundamental concern still remains regarding the role of classical KATP channel in glucose-stimulated insulin release. In the new experiments the author presented (in Fig. 6h, i, j), low dose glibenclamide closes KATP channel, thus depolarizes β -cells and stimulates Ca^{2+} influx. However, unexpectedly, this effect is dependent on SWELL1 expression. The authors speculate that under basal conditions, there is a background of constitutively active SWELL1-mediated depolarizing $I_{Cl,SWELL}$ (at 0 mM glucose). ***This directly contradicts to the data and the literature that SWELL channel needs high glucose levels (i.e. 16 - 20 mM) to be active.*** Only under those conditions, there is presumably enough osmolarity change inside of β -cells."

We agree with the Reviewer that the lack of response of SWELL1 KO β -cells to low dose glibenclamide is a curious finding. Based on the established requirement of SWELL1 for forming the swell-activated chloride channel responsible for VRAC/ $I_{Cl,SWELL}$, we concluded that there must be some basal SWELL1-mediated depolarizing current present in wild-type cells, even at low extracellular glucose concentrations, that is counterbalanced by I_{KATP} . When I_{KATP} is partially blocked with low dose glibenclamide (25-30% we estimate based on published work¹), this basal or background SWELL1-mediated current is sufficient to depolarize the β -cell to threshold for activation. We do not believe that this explanation contradicts our data, nor the literature. Although many reports show that $I_{Cl,SWELL}$ is significantly increased with cell swelling, none show that there is no $I_{Cl,SWELL}$ activity at under basal conditions. On the contrary, there are several published reports of basal $I_{Cl,SWELL}$ /VRAC in neurons^{2,3} and, importantly, in pancreatic β -cells⁴. Specifically, Leonard Best (2002) showed that in cell-attached patches of pancreatic β -cells the frequency of VRAC channel opening is 6-7%⁴ under basal conditions (1 mM glucose). It should also be noted that under conditions of high membrane resistance, as occurs with I_{KATP} blockade, it does not require much depolarizing current to activate the β -cell⁵. Indeed, with full I_{KATP} blockade using high dose glibenclamide even SWELL1 KO pancreatic β -cells⁴ can be stimulated - suggesting the involvement of other ionic currents under these conditions. Our main point is that, based on current knowledge of SWELL1 mediating specifically $I_{Cl,SWELL}$, a

reasonable explanation is that the basal $I_{Cl,SWELL}$ current, shown to be present under basal conditions, may contribute to β -cell depolarization upon I_{KATP} blockade. To clarify this point we have amended the manuscript accordingly to cite the studies that have measured basal $I_{Cl,SWELL}$ under basal, low-glucose conditions (lines 230-239 of revised manuscript, reference numbers are different in the manuscript):

“These data suggests that, under basal conditions, there is a background of constitutively active SWELL1-mediated depolarizing $I_{Cl,SWELL}$ that is balanced by hyperpolarizing I_{KATP} to maintain resting β -cell membrane potential. Indeed, basal $I_{Cl,SWELL}$ has been reported in neurons^{2,3} and, importantly, has been measured in β -cells (NP = ~ 0.06 , at 1 mM glucose)⁴. Near full I_{KATP} inhibition with higher dose glibenclamide (10 nM, 0 mM glucose)¹ is capable of activating intracellular Ca^{2+} in both WT and SWELL1 KO β -cells (Fig. 6k-m), just as robust $I_{Cl,SWELL}$ activation with hypotonic swelling (at 0 mM glucose) can overcome I_{KATP} and trigger Ca^{2+} transients (Fig. 4g) and insulin release from β -cells⁶. Taken together, these data support a model whereby SWELL1-mediated $I_{Cl,SWELL}$ and I_{KATP} counterbalance each other to regulate glucose-stimulated β -cell Ca^{2+} signaling.”

2. “In Fig 6f, g, k, l, m, the authors confirmed that KATP closure is necessary and sufficient for glucose-stimulated β -cell excitation, which is well established in the literature. However, knocking out SWELL1 alone completely abolishes the glucose-stimulated Ca^{2+} entry and insulin secretion as shown in vitro (Figs 4, 5, and 7) and in vivo (Fig. 8). **Is this due to the KATP channel function compromised in SWELL1 KO cells and mice?** If the KATP channel function is intact as suggested by the authors, **should one expect the effects of its closure (at least partially) on glucose-stimulated membrane depolarization, Ca^{2+} entry and insulin secretion?**”

We agree with the Reviewer on the point that one would expect to observe the effects of KATP closure on glucose-stimulated membrane potential depolarization if KATP closure is intact in SWELL1 KO β -cells. Indeed, this is what we observe. As shown in **Figure 3g-i**, SWELL1 KO β -cells do in fact depolarize in response to glucose, and we suspect that this is the component contributed by KATP closure. However, this MP depolarization appears to fall short of threshold for activation of VGCCs, and therefore does not trigger glucose-stimulated Ca^{2+} transients in SWELL1 KO β -cells. In this way, one can imagine the cooperative activity of SWELL1 mediated $I_{Cl,SWELL}$ and KATP as a “two-stage rocket” with KATP closure (first stage) followed by $I_{Cl,SWELL}$ activation (second stage) to drive β -cell MP to threshold for VGCC activation.

3. “The cooperative act of the closure of KATP and opening of other Cl^- channels was also supported by the critical role of several other chloride channels (especially CFTR, Guo JH, Nature Communications 2014, and TMEM16A, Edlund A, BMC Medicine 2014) in modulating glucose-stimulated insulin secretion in β -cells. The authors may want to cite and discuss their findings in this context. In light of these findings, the complete lack of glucose-stimulated Ca^{2+} entry and insulin secretion in SWELL1 KO cells and mice may suggest other yet-unrealized essential functions of VRAC in β -cell biology in addition to the proposed regulation of membrane potential.”

We thank the Reviewer for this point. We have amended our Discussion with the following modified paragraph (lines 353-362, reference numbers are different in the manuscript):

“In addition to SWELL1-mediated $I_{Cl,SWELL}$ there are certainly other depolarizing currents that contribute to β -cell excitability, since full suppression of I_{KATP} is capable of stimulating SWELL1 KO β -cells. These include Transient Receptor Potential (TRP) channels, including TRPM3⁷, TRPM4⁸ and TRPM5⁹⁻¹¹; though not all of these TRP channels are necessarily expressed in human β -cells¹². CFTR has been implicated as Cl^- conductance important for glucose-stimulated β -cell excitability and secretion¹³ as has ANO1/TMEM16A¹⁴, however, the presence of CFTR in the β -cell is controversial¹⁵. Finally, CIC-3, a chloride channel localized to insulin granules, has been proposed to regulate insulin processing and secretion via acidification mechanisms¹⁶. Given the significant impact of SWELL1 on β -cell excitability, insulin secretion and systemic glycaemia it is possible that SWELL1 may have other, as yet unrealized, ion channel regulatory functions in β -cell biology.”

Responses to Reviewer #3:

1. “I understand that swell is not stricto sensu a glucose sensor. But the term “virtual” glucose sensor is confusing. I would suggest to find something else or keep “glucose sensor” as in the previous version but discussing in the paper that it this term should be qualified.”

We thank the Reviewer for this suggestion. We will delete the word “virtual” and use “glucose sensor” instead.

2. “The authors tested the effect of 3-O-methylglucose on $[Ca^{2+}]_i$. They show that it does not increase $[Ca^{2+}]_i$. This is well established. The question was whether a 3-O-methylglucose or blockade of glucose metabolism affects the SWELL1 current.”

We agree with the Reviewer, however the effect of 3-O-methylglucose on β -cell $I_{Cl,SWELL}$, β -cell MP and β -cell volume has been thoroughly established^{4, 17}, which is why we only measured intracellular Ca^{2+} as a read-out in our experiments. Miley et al¹⁷ showed that 16 mM 3-O-methylglucose results in a minor and transient increase in volume (the proposed $I_{Cl,SWELL}$ activator) which is significantly less than the response obtained by equivalent glucose in rat pancreatic β -cells. In addition, these authors demonstrated that 16 mM 3-O-methylglucose has no effect on β -cell membrane potential, which is expected to depolarize upon activation of $I_{Cl,SWELL}$, as we (**Figure 3a-f**) and others¹⁷ observe upon hypotonic activation of $I_{Cl,SWELL}$. Finally, Best actually measured $I_{Cl,SWELL}/VSAC$ in β -cells upon application of 12 mM 3-O-methylglucose and observed no increase in channel activity⁴. Based on these previously published data we did not measure $I_{Cl,SWELL}$ in response to 3-O-methylglucose.

3. “Looking at fig 2, it is not clear at all where a RVD occurs DURING the application of glucose. 15 min after the application of glucose, there is not difference in cell volume between WT and SWELL1 KO and no sign of RVD.”

Figure 2A shows that β -cell volume increases similarly in both WT and SWELL1 KO cells upon glucose application up to around 10 minutes. $I_{Cl,SWELL}$ takes a few minutes to activate since it is activated by β -cell swelling, so it is expected that WT and SWELL1 KO β -cell volume will track each other early on since $I_{Cl,SWELL}$ will not be sufficiently activated until the β -cell actually swells (as shown in Fig 2d). From 10-20 minutes, the SWELL1 KO and WT volumes begin to separate as RVD occurs in the WT β -cells and not in the SWELL1 KO β -cells, particularly when the glucose is switched back from 16.7 mM to 1 mM. After the 20 minute time-point there are clear

differences in β -cell volume in WT compared to SWELL1 KO β -cells that we conclude arises from RVD in WT and not in SWELL1 KO β -cells.

4. “The authors did not fully address the question regarding the KATP channel modulators. Do the modulators affect the SWELL1 current?”

The KATP modulators diazoxide and glibenclamide are thought to be specific to KATP, but we agree that there is a possibility that they may modulate $I_{Cl,SWELL}$ in the reverse fashion to provide an alternative interpretation of the data. To address this we tested KATP channel activator diazoxide (100 μ M) for $I_{Cl,SWELL}$ inhibition (new Supplementary Figure 4a-b). We show a representative trace (from $n = 4$ experiments) to demonstrate that 100 μ M diazoxide does not block $I_{Cl,SWELL}$, and we amended line 228. We also confirmed that glibenclamide does not activate $I_{Cl,SWELL}$ (or any Cl^- currents for that matter) under physiological conditions (new Supplemental Figure 4c-d). We have modified the revised manuscript to indicate this (line 231).

5. “The authors did not answer to one request: **KO of SWELL1 impairs glucose-induced insulin secretion. Does it affect high K induced insulin release?**”

For the mouse and human islet glucose-induced insulin secretion (GSIS) assays, instead of KCl, we use L-arginine plus high concentration (16.7 mM) glucose as positive control for intact insulin secretion in otherwise healthy β -cells. L-arginine, independent of NO, potentiates glucose induced insulin secretion through stimulation of membrane depolarization (L-arginine is a positively charged amino acid, equivalent to positive ions flowing into the β -cell, much like high KCl)¹⁸. Also, L-arginine sensitizes glucose stimulation via two additional mechanisms: by activation of protein kinase A and by activation of protein kinase C (sensitization of the exocytotic machinery to even small changes in Ca^{2+} influx in response to L-arginine)¹⁸. We observe that L-arginine plus 16.7 mM glucose largely enhanced glucose-induced insulin secretion compared to basal secretion level in both WT and SWELL1 deficient islets – confirming that SWELL1 KO/KD islets are viable and capable of secreting insulin similar to WT islets under these conditions. With respect to specifically high K^+ stimulation, it should be noted that all of our Ca^{2+} imaging experiments use KCl stimulation to confirm intact β -cell excitability in WT and SWELL1-deficient β -cells (**Figures 4&5**).

6. “In fig 6b, 6d, the effect of KCl has been tested after removal of bumetanide. It should have been tested during the application of bumetanide.”

In these experiments, KCl was only used confirm β -cell viability and excitability. The KCl stimulus has nothing to do with the bumetanide application. Bumetanide was applied to drop β -cell intracellular Cl^- concentration and convert the normally depolarizing $I_{Cl,SWELL}$ into a non-stimulatory or inhibitory, hyperpolarizing current and revealed the importance of elevated intracellular Cl^- for glucose-induced β -cell excitability.

7. “Fig. 6K, why 10 nM glib increase $[Ca^{2+}]$ with the same delay in WT than in SWELL1 KO cells?”

The exemplar trace shows a similar delay between WT and SWELL1 KO, but overall the time from glibenclamide perfusion to β -cell firing was actually quite variable in these experiments and we never quantified this. According to Gopalakrishnan et al.¹, the IC_{50} of glibenclamide for KATP

channels expressed in HEK-293 cells is 0.92 ± 0.01 nM and IC_{50} for RINm5F β -cell lines is 0.25 ± 0.06 nM. Therefore, 10 nM glibenclamide is a very high concentration, which will inhibit almost all of the KATP channels in β -cells. Under these conditions, with a very high resistance β -cell membrane, we suspect that even in the absence of SWELL1-mediated $I_{Cl,SWELL1}$ (particularly at 0 mM glucose), other tiny background currents are sufficient to ultimately depolarize the β -cell in a stochastic manner.

References

1. Gopalakrishnan, M. *et al.* Pharmacology of human sulphonylurea receptor SUR1 and inward rectifier K(+) channel Kir6.2 combination expressed in HEK-293 cells. *Br J Pharmacol* **129**, 1323-1332 (2000).
2. Zhang, H., Cao, H.J., Kimelberg, H.K. & Zhou, M. Volume regulated anion channel currents of rat hippocampal neurons and their contribution to oxygen-and-glucose deprivation induced neuronal death. *PLoS One* **6**, e16803 (2011).
3. Inoue, H. & Okada, Y. Roles of volume-sensitive chloride channel in excitotoxic neuronal injury. *The Journal of neuroscience : the official journal of the Society for Neuroscience* **27**, 1445-1455 (2007).
4. Best, L. Study of a glucose-activated anion-selective channel in rat pancreatic beta-cells. *Pflugers Arch* **445**, 97-104 (2002).
5. Ashcroft, F.M. & Rorsman, P. K(ATP) channels and islet hormone secretion: new insights and controversies. *Nat Rev Endocrinol* **9**, 660-669 (2013).
6. Best, L., Miley, H.E. & Yates, A.P. Activation of an anion conductance and beta-cell depolarization during hypotonically induced insulin release. *Exp Physiol* **81**, 927-933 (1996).
7. Wagner, T.F. *et al.* Transient receptor potential M3 channels are ionotropic steroid receptors in pancreatic beta cells. *Nature cell biology* **10**, 1421-1430 (2008).
8. Cheng, H. *et al.* TRPM4 controls insulin secretion in pancreatic beta-cells. *Cell Calcium* **41**, 51-61 (2007).
9. Colsoul, B. *et al.* Insulin downregulates the expression of the Ca²⁺-activated nonselective cation channel TRPM5 in pancreatic islets from leptin-deficient mouse models. *Pflugers Arch* **466**, 611-621 (2014).
10. Colsoul, B. *et al.* Loss of high-frequency glucose-induced Ca²⁺ oscillations in pancreatic islets correlates with impaired glucose tolerance in *Trpm5*^{-/-} mice. *Proc Natl Acad Sci U S A* **107**, 5208-5213 (2010).
11. Philippaert, K. *et al.* Steviol glycosides enhance pancreatic beta-cell function and taste sensation by potentiation of TRPM5 channel activity. *Nat Commun* **8**, 14733 (2017).
12. Marabita, F. & Islam, M.S. Expression of Transient Receptor Potential Channels in the Purified Human Pancreatic beta-Cells. *Pancreas* **46**, 97-101 (2017).
13. Guo, J.H. *et al.* Glucose-induced electrical activities and insulin secretion in pancreatic islet beta-cells are modulated by CFTR. *Nat Commun* **5**, 4420 (2014).

14. Edlund, A., Esguerra, J.L., Wendt, A., Flodstrom-Tullberg, M. & Eliasson, L. CFTR and Anoctamin 1 (ANO1) contribute to cAMP amplified exocytosis and insulin secretion in human and murine pancreatic beta-cells. *BMC Med* **12**, 87 (2014).
15. Sun, X. *et al.* CFTR Influences Beta Cell Function and Insulin Secretion Through Non-Cell Autonomous Exocrine-Derived Factors. *Endocrinology* **158**, 3325-3338 (2017).
16. Deriy, L.V. *et al.* The granular chloride channel ClC-3 is permissive for insulin secretion. *Cell metabolism* **10**, 316-323 (2009).
17. Miley, H.E., Sheader, E.A., Brown, P.D. & Best, L. Glucose-induced swelling in rat pancreatic beta-cells. *J Physiol* **504 (Pt 1)**, 191-198 (1997).
18. Thams, P. & Capito, K. L-arginine stimulation of glucose-induced insulin secretion through membrane depolarization and independent of nitric oxide. *Eur J Endocrinol* **140**, 87-93 (1999).

Reviewer #1 (Remarks to the Author):

The authors have satisfied this reviewers questions and comments.
Congratulations

Reviewer #2 (Remarks to the Author):

The authors addressed all my questions. The "two-rocket" model the authors proposed places SWELL1 as central (both necessary and sufficient) to the glucose-stimulated insulin secretion as the classic KATP channel. Since both KATP (channel and its modulator) gain and loss of function directly cause human diseases, the authors may want to discuss the rich human genetics data (GWAS, expression, sequencing) on the potential association of SWELL1 and other VRAC subunits with glucose metabolic diseases.

Reviewer #3 (Remarks to the Author):

The authors answered to all concerns